# Context-dependent interaction between goal-directed and habitual control under time pressure
Ben J. Wagner [1,2,5], Sascha Frölich [1,5], Sarah Schwöbel[1], Michael N. Smolka [3] &
Stefan J. Kiebel [1,4] ✉

Current theories of habitual behaviour assume that habits and goal-directed actions are controlled by two separate but interacting systems. However, it is not clear how these systems interact when actions must be made under time pressure. We use a task that induces habitual action sequences while requiring participants to perform goal-directed actions that are either congruent or incongruent with the habit. This task allows for the concurrent measurement of goal-directed and habitual behaviour, permitting a nuanced analysis of the interaction between these two control modes in a sample of 60 participants (30 male and 30 female). Using computational modelling, we find that models in which habit influence depends on the number of repetitions explain participant behaviour better than models assuming constant habit strength. We further show that participants modulate their use of the habit depending on the context: they selectively inhibited the habit when it was incongruent with their explicit goals and exploited it when the two were congruent. Drift-diffusion modelling of choices and reaction times shows a shift in response-bias when habits and goals align. Together, our results indicate that habitual and goal-directed actions interact in a context-dependent manner allowing for the adaptive allocation of proactive and reactive control.

Habits are an essential aspect of human behaviour. Some studies estimate that a third to one half of human behaviour is habitual[1]. Habits are fast and reflexive, and allow us to efficiently select actions that are repeated often within the same context[2]. Because of their significance and influence on behaviour, habits have been an active field of research in psychology and neuroscience for decades. Moreover, maladaptive habits represent pivotal mechanisms underlying a range of mental disorders, including obsessive-compulsive disorder and addiction[3,4].

One hallmark of habitual behaviour is that it is faster than goal-directed behaviour[5]. Hardwick et al.[6] show that habitual behaviour dominates action selection when actions have to be performed fast, and is replaced by goal-directed behaviour when enough time is available. Classical habit theory further posits habits as stimulus-response (S-R) associations that are initially executed as part of a stimulus-response-outcome (S-R-O) chain[7–9]. In this view, an agent first tries to obtain a given outcome (O) by calculating and performing a goal-directed response (R) in presence of a given stimulus (S). With extensive repetition and reinforcement, the S-R associations become strengthened, until the stimulus alone triggers the habitual response[7]. While

S-R learning is able to explain many experimental findings, it has also become clear that acquisition and expression of habits strongly depend on the given context[2,9].

In computational approaches to behaviour, habitual and goal-directed control are usually modelled as two distinct controllers[10–15]. This seems in line with earlier neuroscientific evidence showing that habitual and goal-directed behaviour are subserved by distinct corticostriatal connections[7,8]. However, recent work suggests that habitual and goal-directed processing are neurally intertwined rather than cleanly separable[16], supporting models that emphasize overlapping and interacting corticostriatal circuits[17] as well as arbitration and control mechanisms[16,18,19], rather than a clear separation. Likewise, some of these computational approaches assume the output of the two controllers to be mixed or arbitrated during behavioural selection[10–13,20,21]. In a mixing model, both the habitual and goal-directed controllers are assumed to compute their outputs in parallel. This implies that the goal-directed controller is always engaged, which makes it difficult to explain why habitual actions are executed faster than goal-directed ones. Other studies have described different types of interactions between

[1]Department of Psychology, Technische Universität Dresden, Dresden, Germany. [2]Department of Psychiatry and Psychotherapy, University of Tuebingen, Tuebingen, Germany. [3]Department of Psychiatry and Psychotherapy, Technische Universität Dresden, Dresden, Germany. [4]Centre for Tactile Internet with Human-in-the-Loop, Dresden, Germany. [5]These authors contributed equally: Ben J. Wagner, Sascha Frölich. ✉e-mail: stefan.kiebel@tu-dresden.de

goal-directed and habitual control[14,15,22–24]. For example, in ref. 23, habits are modelled as action sequences, where the goal-directed controller selects an individual action, or a habitual action sequence. Congruent with experimental findings, the execution of the action sequence is characterised by fast and inflexible performance of the individual constituent actions.

Another option for computational modelling, which has so far received far less attention, is to harness the experimentally established context-specificity of habits[2,9,21]. The idea here is to make the interaction between goal-directed and habitual controllers explicitly context-dependent e.g., ref. 25. In such a framework, an agent infers the current context to apply a previously learned context-specific arbitration between goal-directed and habitual behaviour. For example, when in the context of standing in the bathroom in the morning, the brain infers the context of 'I just got up, walked into the bathroom and need to prepare for the day' and starts the habit stored in that context. Later during the day, when in the bathroom again at a different time, the brain infers a different context and brushing one's teeth is not a habit but just one possible option. Such a simple but rapid context-specific arbitration moves the heavy lifting of the underlying computations from an arbitration between controllers to the inference of context. This may explain how the brain can minimize the computations required for the assumed arbitration between goal-directed and habitual control, especially under demanding response time deadlines[26].

However, despite these theoretical advances, there is still limited empirical evidence on how quickly and flexibly habitual and goal-directed systems interact depending on contextual demands, particularly under time pressure. Accordingly, the present study aimed to test for a context-specific and rapid interaction of habitual and goal-directed behaviour. To do this, we used a revised and refined variant of the Action Sequence Task (AST), which we have used previously[27], and which is built on the serial reaction time task[28–30]. In the AST, participants extensively repeat an action sequence under time pressure, since participants are instructed to respond within a tight deadline of a few hundred milliseconds (e.g., 600 ms). Through repetition, and probably due to the need for fast responses, these action sequences become automatic quickly[27,30]. Sequences are executed mostly without explicit awareness of a sequential pattern.

Crucially, in the AST, common trials in which the execution of the learned action sequence was beneficial were sometimes followed by similarly appearing trials which required a goal-directed response. We implemented two different contexts of such trials. In the congruent context, the learned habitual action was the same as the goal-directed response (goal-directed action is congruent with habitual action), while in the incongruent trial context, the habitual action was different from the goal-directed response (goal-directed action is incongruent with habitual action). This relates to classical accounts of habit formation in two ways. First, repeated execution of an action or a sequence of actions strengthens performance, consistent with theories that conceptualise habits as learned action sequences[23,31]. Second, the design allows us to test whether the learned sequence continues to influence behaviour when they no longer lead to high-reward outcomes, i.e., whether habitual responding persists under devaluation of the action-sequence, i.e., when the habitual and goal-directed responses conflict. In consequence, we do test whether participants learn to modify the strength of the habitual action in a context-specific fashion.

To investigate the role of repetition on choice behaviour, we modelled choice data by comparing two model families. The first model family explained choices by the influence of a reward-learning goal-directed controller, and a value-free repetition-learning based habitual controller. The goal-directed controller learned $Q$-values values of actions using temporal-difference learning based on received reward similar to ref. 13. The habitual controller learned a repetition-value for each action based on action sequences a participant has performed in the past, building on the concept of habits as action sequences[23] and on the idea that habits are driven by repetition[21,25]. In the second model family, behaviour was purely based on reward-learning using temporal-difference learning of $Q$-values. Within each model family, we further differentiated between models where the influence of the habit is equally strong in all trials that required a goal-directed response, and where that influence is modulated depending on the congruency of the habit with the participants' explicit task goals (that is, dependent on the context).

In addition, we used drift-diffusion modelling (DDM) to model both choice and reaction time data simultaneously[32,33]. Specifically, we aimed to investigate whether the learned habitual action sequences impact participants' prior response tendencies (proactive control), or whether they affect participants' choices at the level of information processing, that is, evidence accumulation (reactive control). Results from previous studies have been mixed as to whether prior information, before entering a trial, mainly affects response bias[34,35], or whether the influence of prior information modulates both response bias and drift rate, depending on the task at hand[36]. Recently[37], proposed a sequential sampling model in which habitual response tendencies map onto the response bias, whereas goal-directed responses influence the drift rate. Here, we test whether learned habitual action sequences influence prior response tendencies by modulating the DDM's starting point bias, or whether they influence the decision-making process itself by modulating evidence accumulation.

## Methods
### Participants
We recruited participants on the online platform prolific.com. Exclusion criteria for study registration were age (below 25 or above 60 years), current behavioural or pharmacological treatment of a mental disorder, regular intake of psychotropic medication, impairments that make the use of a computer keyboard hard or impossible, and substance use disorders (except tobacco). Participants were provided informed consent prior to participation and then performed the experiment online. Between instructions and the main experiment participants performed on a criterion test, excluding approximately 14% of participants (see supplementary material for details). The high fail rate is likely in part due to the online nature of the experiment. Of 81 participants who completed the experiment, 16 were excluded for experimental reasons (see supplementary material), resulting in 65 eligible participants (16 in task version 1, 15 in task version 2, 18 in task version 3, and 16 in task version 4). We apriori decided to obtain a balanced sample per task set-up (ordering of repeating-sequence and random blocks, reward locations) and therefore because group with task version 2 had the lowest number of participants (15), we balanced the final sample by including exactly 15 participants per group, resulting in the 60 participants (30 male and 30 female) reported in the main manuscript. Information on participants gender was provided by participants. For transparency, we re-ran our main analyses on the full sample of 65 participants and report these in the supplementary information. Of the 60 participants, 30 were male and 30 female. Average age was 37.7 ± 9.03 years (min 25 years, max 58 years). The study was approved by the Ethics Commission of TU Dresden (EK 514122018) and was not preregistered. Data on race and/or ethnicity was not collected.

### Task
We used a revised and refined version of the Action-Sequences Task (Frölich et al.[27]; Fig. 1; see also supplementary material). Participants saw one or two stimuli appear in four possible locations on the computer screen (Fig. 1A). Each location was mapped to one of four keys on the computer keyboard (s,x,k, and m). Participants had 600 ms to respond to the appearance of either a single stimulus (single-target trial; STT) or two stimuli (dual-target trial; DTT) by pressing the corresponding key. In DTTs, participants were free to choose which of the two mapped keys to press, while one of them was associated with a low reward probability (20%) and the other with a high reward probability (80%). The task primarily consisted of STTs, which were occasionally interspersed with DTTs (on average one DTT every six to seven STTs). Figure 1A illustrates an example sequence of trials, showing the transition from an STT to a DTT (from left to right). If participants responded correctly within the response deadline, they were probabilistically rewarded with a point. Reward feedback was provided visually: a green smiley was displayed in

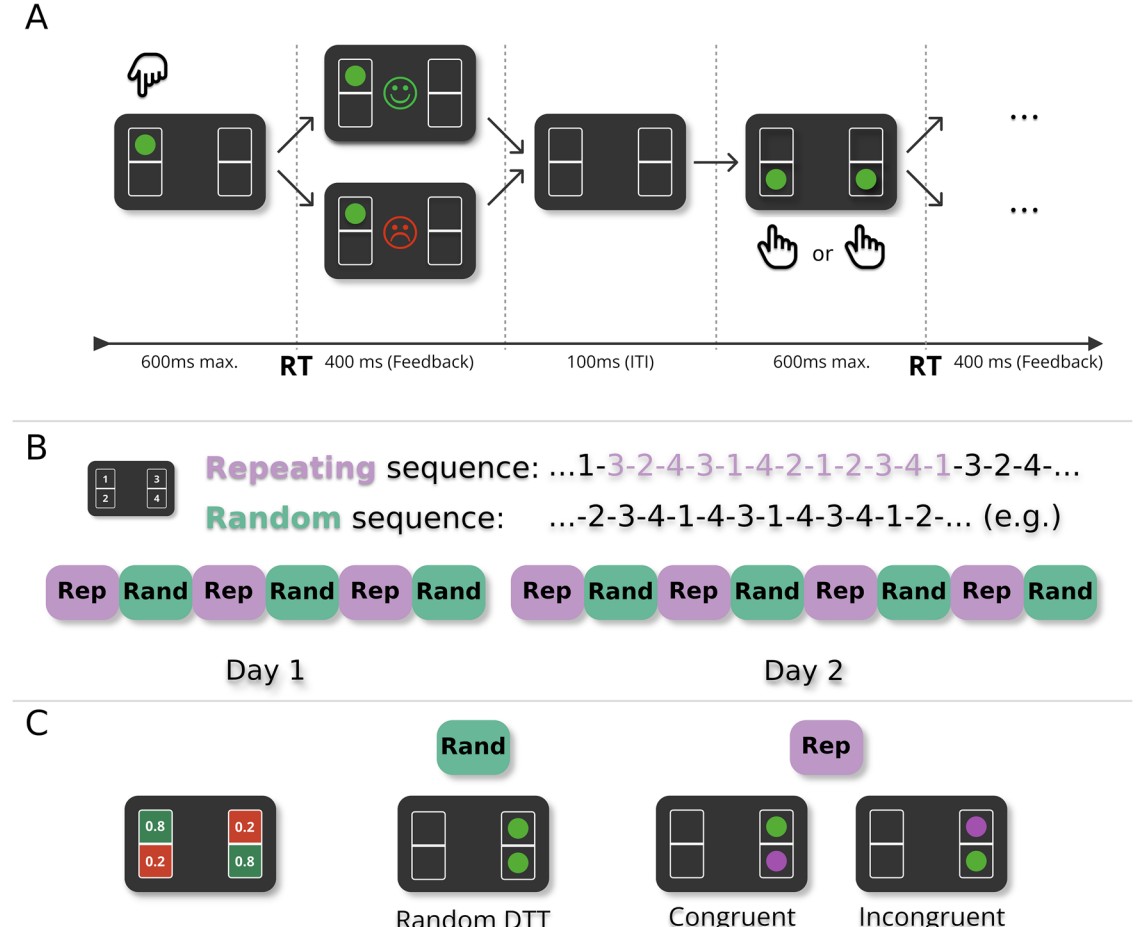

**Fig. 1 | Experimental design. A** In each trial, participants saw one or two green target stimuli placed in four different boxes. Participants had to press one of four corresponding keys on the keyboard in response to the target position(s), within a deadline of 600 ms after stimulus onset. When pressing the correct key, a point reward was indicated probabilistically. If the trial was rewarded, a smiley was shown in the centre of the screen. In the case of no reward, a frowny appeared instead. This feedback phase lasted 400 ms. After the feedback, the intertrial interval (ITI) lasted 100 ms. **B** There were two experimental conditions: In the *repeating-sequence* condition (Rep), stimuli kept repeating the same twelve-item sequence (here purple for illustration), while in the *random-sequence* condition (Rand), targets appeared pseudo-randomly. The experiment was performed over the course of two consecutive days with alternating sequential and random blocks, six blocks on day 1 and eight blocks on day 2. Each block consisted of 480 trials. Block order was counterbalanced (see supplementary material). **C** Two of the four stimulus positions were associated with a high reward probability of 0.8. The other two stimuli were associated with a low reward probability of 0.2. reward probabilities did not change over the course of the experiment and the positions of high and low reward probability were counterbalanced (see supplementary material). dual-target trials (two green targets) appeared pseudo-randomly, with a frequency of 15% of all trials, i.e., 72 dual-target trials per block. In the repeating-sequence (Rep) condition, one of the two targets always corresponded to the current sequence element (here purple for illustration). In the repeating-sequence condition, dual-target trials (DTT) were either *congruent* if the sequence element was in a location with high reward probability (7.5% of all trials), or *incongruent* if the sequence element was in a location with low reward probability (7.5% of all trials).

the centre of the screen for rewarded trials, and a red frowny was shown when no reward was obtained.

In the case of errors, i.e., timeouts or double key presses, a penalty screen appeared for 1200 ms with a corresponding text (either "Too slow", or "Please only press one key at a time"). Each condition (Rep or Rand) consisted of 480 trials (408 single-target trials and 72 dual-target trials) (Fig. 1B). In the Rep condition, stimuli were presented as a repeating sequence of twelve stimulus positions. In the Rand condition, stimuli appeared pseudo-randomly and did not follow a repeating sequence. During instruction, participants were explicitly told which two locations had the highest reward probabilities, but were not told the exact values (80% and 20%, respectively). The assignment of high- and low-reward locations remained fixed for each participant (counterbalanced) across both days. Each DTT contained one stimulus with low reward probability, and one stimulus with high reward probability. Crucially, in some DTTs of the Rep condition, the explicit instruction of choosing the higher reward probability conflicted with the repeating stimulus sequence. Specifically, a DTT of the Rep condition was considered *congruent* if the sequence element coincided with the location of the higher reward probability. Conversely, it was considered *incongruent* if the sequence element was in the location with lower reward probability. Participants were not told about the existence of the Rep and the Rand conditions, and were instructed to collect as many points as possible. They were told that their monetary reward at the end of the experiment depended on the total number of collected points. The experiment was performed over the course of two consecutive days, and participants had to perform both parts of the experiments around the same time of day (within the same four-hour time window). See the supplementary methods for more details, including a description of changes of the task since[27] Table 1.

**Computational modelling**

**Generative modelling of choice behaviour.** Computational modelling allows us to test hypothesised cognitive mechanisms involved in response selection, and measure the strength of those effects by making use of within-subject response variances. We modelled responses in dual-target trials (DTT) to test whether responses in DTT are influenced by the

## Table 1 | Glossary of terms

| Term | Meaning |
|---|---|
| STT | single-target trial. Only one stimulus is shown. Participants were instructed to respond with the corresponding button press. 85% of trials were single-target trials. |
| DTT | dual-target trial. Two stimuli are shown. Participants were instructed to respond with the press of one of the two corresponding keys. Participants were instructed to try and choose the one with higher reward probability. Each DTT contained exactly one stimulus with high reward probability (80%) and exactly one with low reward probability (20%). 15% of trials were dual-target trials. |
| congruent DTT | A dual-target trial within the Rep condition (see below), where the sequence element is associated with high reward probability. |
| incongruent DTT | A dual-target trial within the Rep condition (see below), where the sequence element is associated with low reward probability. |
| congruent context | The context of a congruent DTT, i.e., a trial when goal-directed and habitual controllers require the same action (no response conflict). |
| incongruent context | The context of an incongruent DTT, i.e., a trial when goal-directed and habitual controllers require different actions (response conflict). |
| random DTT | A dual-target trial within the Rand condition (see below). |
| Rep | Repeating-sequence condition. A task condition, where each trial (STT or DTT) showed an element of a repeating 12-item sequence. |
| Rand | Random-sequence condition. Here, stimuli of STTs and DTTs appeared in a pseudo-random positions. |
| Optimal Choice | Choosing the response option with high reward probability in a DTT. |

action sequence in the preceding single trial targets (STT), and whether there is evidence for a dynamical adaptation of habitual responding, depending on the type of DTT (i.e., congruent or incongruent context, see Glossary). By design, the habitual and contextual components are computationally separable. The habitual component reflects a repetition-based action sequence learned from past sequence executions, whereas the contextual component is defined by the alignment between this habitual tendency and the goal-directed optimal action (high-reward location) on a given dual target trial (congruent vs. incongruent) or the absence of a sequence (random context). We compared six candidate models from two model families, one of which implements context-dependent modulation of the habitual component via a dedicated parameter. Model fitting was performed with hierarchical stochastic variational inference in pyro 1.8.6 with Python 3.11.6 and within RStudio 2023.12.1 and JAGS 4.3.2 followed by formal model comparison.

We compared models of two model families. Model Family 1 consists of models with two distinct action controllers, a temporal-difference learning-based goal-directed action controller, and a repetition-based habit controller. Models in Model Family 2 were modelled analogously to the models in Model Family 1, but do not contain a repetition-based habitual controller. Instead, their behaviour is controlled by temporal-difference learning of Q-values, weighting the Q-values differently depending on the type of dual-target trial.

**Model Family 1: Models with Goal-Directed and Habitual Action Controllers.** Model Family 1 modelled choice behaviour as the result of the interaction between a goal-directed behavioural controller and a repetition-based habitual controller. The goal-directed controller was modelled as a simple temporal-difference learning algorithm for Q-value learning, $Q_{t+1} = Q_t + \alpha(O_t - Q_t)$, . Here $O_t$ is the outcome (0 or 1) at trial $t$, and $\alpha$ a free learning rate parameter. One Q-value was learned for each of the four possible stimulus positions. Note that although participants were informed that some locations were generally associated with higher versus lower reward likelihood, they were not told the exact reward probabilities. Temporal-difference learning, therefore, captures how reward contingencies are learned over time and how the resulting value estimates are used to guide choices on DTTs. The habitual controller was modelled based on performed action sequences. Each action $a$ in addition to a goal-directed Q-value was associated with a repetition-bias $R(a)$. This value was computed as the ratio of the number of times action sequence $a_{t-2}, a_{t-1}, a_t = a$ was performed since the beginning of the experiment, to the number of all performed action sequences beginning with $a_{t-2}, a_{t-1}$:

$$R(a = a_i) = \frac{\#(a_{t-2}, a_{t-1}, a_t = a_i)}{\sum_{j=1}^{4} \#(a_{t-2}, a_{t-1}, a_t = a_j)}, \quad (1)$$

where $i, j \in \{1, 2, 3, 4\}$. All actions in STTs and DTTs were considered for the computation of the repetition-bias. The counts of the previously performed action sequences $\#(a_{t-3}, a_{t-2}, a_{t-1}, a_t)$ were tallied separately for the Rand and the Rep condition.

Model 1 (3 parameters): Simple repetition-bias model. This model computes response options based on learned Q-values and repetition-bias values $R(a)$ and assumes no qualitative difference between congruent and incongruent DTTs. The probability of a goal-directed response is modelled as a sigmoid choice rule:

$$p(a_{opt}) = \sigma(\theta_Q \Delta Q + \theta_{Rep} \Delta R), \text{ where} \quad (2)$$

$$\Delta Q = Q(a_{opt}) - Q(a_{subopt}), \text{ and} \quad (3)$$

$$\Delta R = R(a_{opt}) - R(a_{subopt}) \quad (4)$$

$\theta_Q$ and $\theta_{Rep}$ are free model parameters. We call the response option with higher reward probability the optimal response ($a_{opt}$), and the one with low reward-probability the suboptimal response option ($a_{subopt}$). Note that, Q-values for $a_{opt}$ do not always have to be higher than those for $a_{subopt}$. After learning however, $Q(a_{opt}) > Q(a_{subopt})$, and thus $\Delta Q > 0$. In congruent DTTs, $a_{opt}$ corresponds to the repeating action sequence, while in incongruent DTTs, $a_{subopt}$ corresponds to the repeating action sequence. After learning, the difference between the response-bias values for the two actions, $\Delta R$, is positive in the case of congruent DTTs, and negative in incongruent DTTs and thus intrinsically differentiates between congruent and incongruent contexts. $\theta_Q$ and $\theta_{Rep}$ are fitted across the Rep and the Rand conditions (see Fig. 1b). Note that all model parameters were fitted separately for days 1 and 2.

Model 2 (4 free parameters): Repetition-bias model with context-switch I. This model is an extension of the simple repetition-bias model (model 1, Eq. (3)) and formalizes the assumption that a conflict between goal-directed and habitual controller modulates the influence of the habitual action sequence, using an additional term that allows for a dynamic adaptation of the repetition-bias, depending on the DTT context (congruent or incongruent):

$$p(a_{opt}) = \sigma(\theta_Q \Delta Q + (\theta_{Rep} - \eta_{Inc}\theta_{Inc})\Delta R), \quad (5)$$

where $\eta_{Inc}$ indicates an inferred incongruent context in a DTT. We approximate this inference by setting $\eta_{Inc}$ only to 1 if $\Delta R$ and $\Delta Q$ (see Eqs. (4) and (3)) are of opposite sign, and otherwise to 0. Hence, the influence of the habitual action sequence on choice behaviour is $\theta_{Rep}\Delta R$ in congruent DTTs, and can be modulated as $(\theta_{Rep} - \eta\theta_{Inc})\Delta R$ in incongruent DTTs.

Model 3 (4 free parameters): Repetition-bias model with context-switch II. The previous model assumes a constant change of the effect of the habitual action sequence in case of an incongruent DTT, namely a reduction of $\theta_{Rep}$ by $\theta_{Inc}$ (Eq. (5)). However, it is reasonable to assume that conflict varies over trials. To implement this, we assumed that if at least one of the two terms $\Delta Q$ and $\Delta R$ is small, the conflict between goal-directed controller and habitual controller will be small as well. If the reduction of the effect of the habitual action sequence is triggered by a conflict between the goal-directed and the habitual controllers, this conflict is likely large when both $|\Delta Q|$ and $|\Delta R|$ are large, but small when one of them is almost zero, since in such a case the controller with the small value has no strong preference for one response option over the other. We can account for such a dynamic with the switch-term term $\min\{|\Delta R|, |\Delta Q|\}$:

$$p(a_{opt}) = \sigma(\theta_Q \Delta Q + \theta_{Rep} \Delta R + \eta_{Inc} \theta_{Inc} \min\{|\Delta R|, |\Delta Q|\}). \quad (6)$$

Again, $\eta_{Inc}$ is only 1 in inferred incongruent DTTs, see the description of the Context-Switch model above.

Model Family 2: Pure Q-Value-Learning Models. To test for the presence of the repetition-bias $R(a)$, we fitted models that are analogous to the models in Model Family 1, but without the mechanisms based on repetition-bias $R(a)$. These models predict responses purely by weighting the learned Q-values depending on the current DTT.

Model 4 (3 free parameters): Q-weighting model with identical effect strengths for congruent and incongruent trials. This model learns $\theta_Q$ like the simple repetition-bias model (model 1). Instead of a repetition-bias weighted by $\theta_{Rep}$, this model learns an additional term $\theta_{Qdiff} \Delta Q$, which is added in a congruent DTT and subtracted in an incongruent DTT. Conceptually, $\theta_Q \Delta Q$ should thus encode the effect of the goal-directed controller, while $\theta_{Qdiff} \Delta Q$ encodes the supporting (congruent), or hindering (incongruent) influence of the habitual action sequence in goal-directed responding:

$$p(a_{opt}) = \sigma((\theta_Q + \eta \theta_{Qdiff}) \Delta Q) \quad (7)$$

Since this model is not endowed with a repetition-bias $R(a)$, we have to directly provide it with knowledge of the DTT type ($\eta$): $\eta$ is 1 in congruent DTTs, $-1$ in incongruent DTTs, and 0 in the Rand condition.

Model 5 (4 free parameters): Q-weighting model with context-switch. This model is similar to model 4, but with a qualitative distinction between congruent and incongruent context. In model 4, the deviation of goal-directed responding from random DTTs is of size $\theta_{Qdiff}$ ($\theta_{Qdiff} \Delta Q$), albeit with opposite directions for congruent and incongruent DTTs. Model 5 allows for different sizes of those deviations for congruent and for incongruent DTTs, which makes it analogous to models 2 and 3:

$$p(a_{opt}) = \sigma((\theta_Q + \eta_{Rep} \theta_{Cong} - \eta_{Inc} \theta_{Inc}) \Delta Q). \quad (8)$$

Here $\theta_{Cong}$ encodes the difference in optimal responding between the congruent context and random DTTs, and $\theta_{Inc}$ the modulation in the incongruent context. Similar to Model 4, to enable the model to differentiate between the different DTT types, we set $\eta_{Rep}$ to 1 in DTTs of the Rep condition and $\eta_{Inc}$ to 1 only in incongruent DTTs.

Model 6 (4 parameters): Q-weighting model with independent weights for all DTT types. As in Model 5, this model discriminates between all three DTT types, and predicts responses by weighting Q-value differences with a different parameter for each DTT type:

$$p(a_{opt}) = \sigma(\theta_{Q,i} \Delta Q). \quad (9)$$

$\theta_{Q,i}$ is one of three different model parameters $\theta_Q$, depending on the DTT type, where information about the DTT type is again explicitly given to the model. Theoretically, of all other models, this model should be able to best track the mean value of optimal responses for all DTT types.

**Posterior predictive checks.** We performed posterior predictive checks to assess whether the best-fitting model reproduced key empirical patterns in goal-directed responding across task contexts. We therefore used posterior draws from the fitted hierarchical model to simulate binary goal-directed choices on DTTs for each participant. We then aggregated simulated choices in the same way as the observed data, computing the proportion of optimal (goal-directed) choices separately for each DTT type (random, congruent, incongruent) and directly compared them to the observed group-level proportions.

**Parameter recovery.** We performed parameter recovery for the best-fitting model (model 3). During fitting, we generated posterior-predictive datasets by simulating DTT choice datasets conditional on joint post-warm-up parameter draws, across the whole posterior distribution while preserving the original trial structure, outcomes, and all regressors (subject-level learning rate (lr) and weights $\theta_Q$, $\theta_{Rep}$, $\theta_{Inc}$, each estimated separately for Day 1 and Day 2). Across this posterior-predictive pool, we randomly sampled K = 15 synthetic datasets and refit each dataset with the same hierarchical implementation as in the main analysis. Parameter recovery was assessed by computing Pearson correlations between the known generating parameter values (from the joint posterior draw used to simulate each dataset) and the recovered parameter estimates (posterior means), computed separately for each parameter and day and pooled across subjects and simulated datasets.

**Drift-diffusion modelling.** The previous models with softmax-decision rules for choice behaviour showed that existence of a repetition-bias of the habitual controller, and that around half of participants attenuated the influence of the habitual controller in the incongruent context compared to the congruent context. Those models give however no insight into the cognitive processes that underlie habitual responding in the congruent context and response inhibition in the incongruent context. For example, attenuated habitual responding in the incongruent context could be the result of a reduced prior response tendency for the habitual response, or of a cognitive process that involves information processing, like context inference. Furthermore, the differences in reaction times in congruent, incongruent, and random DTTs suggest that in each of those trial types, slightly different processes are at play. We therefore modelled choice data and reaction times jointly in a drift-diffusion model with collapsing decision boundaries. In contrast to the six models above with softmax choice-rule, the DDM can break down choice behaviour into different cognitive processes, including evidence accumulation and prior response tendency[38,39] (see Fig. 2 for an illustration of a DDM with collapsing boundaries). We examined whether the influence of the habitual action sequence on behaviour manifests at the level of evidence accumulation (drift rate) or whether habitual responses only affect the response bias (starting point). In the first case, the habit influences the choice process itself, while in the second case, it only impacts the prior response tendency. Implementation was done using the HSSM toolbox in Python[40].

Here, we defined the upper response boundary as the optimal response option (80% reward probability), whereas the lower boundary was defined as the suboptimal response option (20% reward probability). We furthermore used an absolute cut-off, such that for each participant, response times below 150ms were excluded from the analysis. Collapsing boundaries simulate time pressure (Fig. 2). The parameter $a$ is the boundary separation (i.e., the amount of evidence required before committing to a decision), $\tau$ the non-decision time (i.e., components of the RT not related to evidence accumulation, like motor preparation), $z$ the starting point of the evidence accumulation process (i.e., a bias towards one of the response boundaries, with $z > 0.5$ reflecting a bias towards the optimal response boundary, and $z < 0.5$ reflecting a bias towards the suboptimal boundary), and $\nu$ the rate of evidence accumulation (the drift rate). Given that the standard DDM likelihood has a long

tail and is poorly suited for modelling data with strict time constraints, the decision threshold $a$ was allowed to collapse within a trial via the angle parameter $\theta$.

## Results

We first describe results obtained by standard linear inference tools, such as ANOVA, t-tests, and Pearson correlations. In the second part, we describe the results obtained by computational modelling.

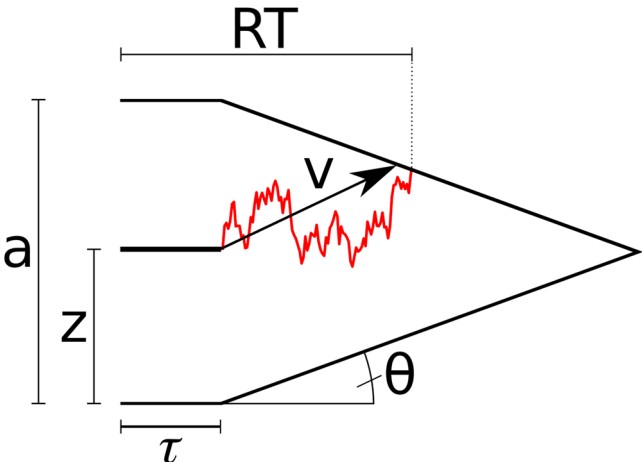

**Fig. 2 | illustration of a DDM with collapsing decision boundaries.** $a$ Boundary separation, $z$ starting point, $\tau$ non-decision time, $v$ drift rate, $\theta$ boundary angle, $RT$ Reaction Time. The upper and lower response boundaries represent the two possible response options in a forced alternative-choice task (like a DTT). The model assumes that a response for the upper (lower) response option is made as soon as the evidence accumulation process, which is formalised as a random walk, reaches the corresponding response boundary. Red illustrates one possible evidence accumulation trajectory.

### Behavioural analysis

**Optimal responding and influence of action sequence.** Behavioural analysis was performed similar to our first study with a previous version of the AST, and results replicate our initially reported findings[27]. A paired two-sample t-test between reaction times in STTs of the Rand and Rep conditions revealed a highly significant difference ($t(59) = 11.0$, $p < 0.0001$, 95% CI of difference in RT [8.31, 12.0], Cohen's $d = 1.43$) (Fig. 3A). Participants responded faster in the Rep condition than in the Rand condition, with an average $\Delta RT = 10.1$ ms. Furthermore, a paired two-sample t-test of error rates between the Rand and Rep conditions also showed a highly significant difference ($t(59) = 6.17$, $p < 0.0001$, 95% CI of difference in error rates: [0.48, 0.93], Cohen's $d = 0.80$), with fewer errors in the Rep condition (Fig. 3B), and an average $\Delta ER = 0.70\%$.

We tested the relationship between the differences $\Delta RT$ and $\Delta ER$ (which are the differences in reaction times (error rates) of STTs between Rep condition and Rand condition, Fig. 3A and B) and the difference in optimal responding in congruent and incongruent DTTs (we denote this difference as $C - I$). Since $\Delta RT$ and $\Delta ER$ measure the strength of the habit outside of choice situations, a positive correlation between those measures and the difference of optimal responding between congruent and incongruent DTTs ($C - I$) indicates that they serve as positive measures of habit (as described in ref. 27). Indeed, we find such a significant positive correlation for both $\Delta RT$ (Pearson $r = 0.65$, $p < 0.0001$, 95% CI of $r$: [0.48, 0.78]) and $\Delta ER$ (Pearson $r = 0.34$, $p = 0.0087$, 95% CI of $r$: [0.09, 0.54]) (Fig. 4).

For each DTT type (congruent, incongruent, random), we computed the proportion of optimal responses (choices of the response option with high reward probability) for each participant as the ratio of high-probability reward responses to all valid responses. A response was counted as valid if participants pressed the key of one (and only one) of the presented stimuli within 600 ms. Average optimal responding across participants and all DTTs was 84.7% (Fig. 3C), showing that participants were able to do the task and generally acted in a goal-directed manner. Note that optimal responding cannot necessarily be interpreted as goal-directed responding in all DTTs: In congruent DTTs, an optimal response might arise from habitual or from goal-directed responding, while in random and

**Fig. 3 | Evidence for reward contingency learning and habitual motor sequence ($n = 60$). A** In single-target trials (STT) of the repeating-sequence condition (Rep), participants displayed shorter reaction times than in the random (Rand) condition. **B** In STTs of the Rep condition, participants performed less errors than in the Rand condition. **C** Proportion of optimal responses for dual-target trial (DTT) types *random*, *congruent* and *incongruent*. In all three conditions, participants responded optimally above-chance levels. The significant difference between congruent and incongruent DTTs indicates that participants learned a repetition bias. **D** Reaction times are different between all three types of DTTs. Error bars indicate standard errors of the mean.

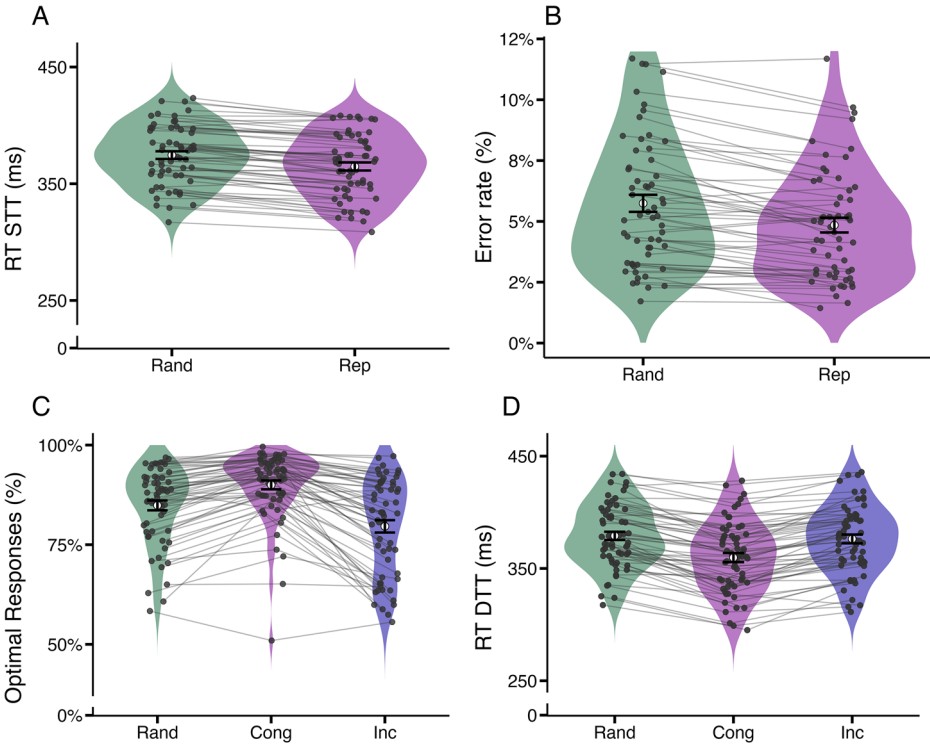

**Fig. 4 | Behavioural differences in single-target trials predict choice differences in dual-target trials ($n$ = 60). A** Reaction-time differences in STTs ($\Delta$RT) correlate with the difference in optimal responding between congruent and incongruent DTTs. **B** Error rate differences in STTs ($\Delta$ER) correlate with the difference in optimal responding between congruent and incongruent DTTs. These results show that reaction-time differences and error-rate differences in STTs induced by the repeating action sequence are predictive of choice behaviour in DTTs. ***:$p < 0.001$, **:$p < 0.01$. The results reported here replicate results reported in ref. 27.

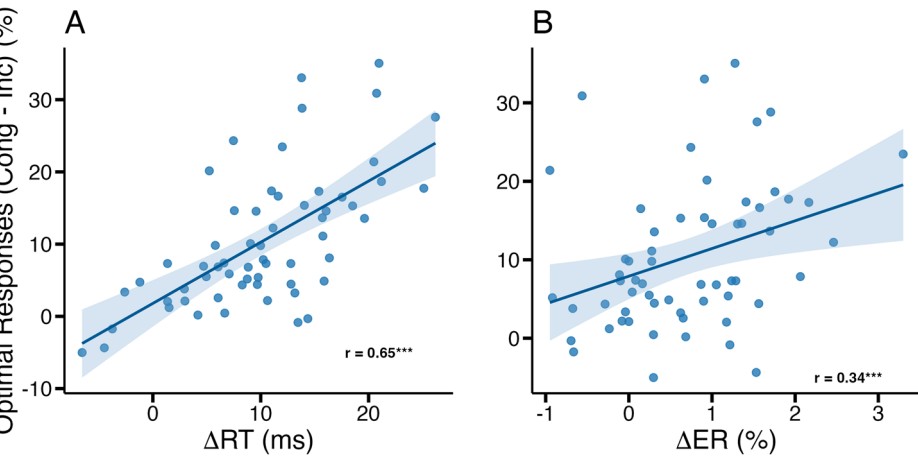

incongruent DTTs, optimal responding can be considered equivalent to goal-directed responding. A repeated-measures ANOVA was performed to analyse the effect of DTT type on optimal responding in DTTs (Fig. 3C). The results indicate a highly significant main effect of DTT type ($F(1.29, 76.21) = 65.3$, $p < 0.0001$, $\eta^2_G = 0.134$, partial $\eta^2 = 0.53$, 90% CI [0.43, 1.00], Greenhouse-Geisser corrected). Paired t-tests of logit-transformed optimal response proportions demonstrated significant differences between all DTT types: congruent responding was significantly higher than both incongruent ($t(59) = 11.1$, $p_{adj} < 0.0001$, 95% CI: [0.79, 1.14], Cohen's $d = 1.44$, 95% CI [1.07, 1.79]) and random DTTs ($t(59) = 9.50$, $p_{adj} < 0.0001$, 95% CI: [0.47, 0.72], Cohen's $d = 1.23$, 95% CI [0.89, 1.56]), while responding in incongruent DTTs was significantly lower than in random DTTs ($t(59) = -7.28$, $p_{adj} < 0.0001$, 95% CI [$-0.46$, $-0.26$], Cohen's $d = 0.94$, 95% CI [0.63, 1.24]) (Fig. 3C). These results show that the underlying action sequence influences choice behaviour, increasing optimal responding in congruent DTTs and reducing it in incongruent DTTs, compared to random DTTs. A repeated-measures ANOVA was next performed to analyse the effect of DTT type on reaction times (Fig. 3D). We found a highly significant main effect of DTT type ($F(1.61, 94.91) = 108.1$, $p < 0.0001$, $\eta^2_G = 0.072$, partial $\eta^2 = 0.65$, 90% CI [0.58, 1.00], Greenhouse-Geisser corrected). Paired two-sample t-tests revealed significant pairwise differences between all DTT types: participants were faster in congruent DTTs compared to both incongruent ($t(59) = -10.3$, $p_{adj} < 0.0001$, 95% CI of difference in RT: [$-19.9$, $-13.5$] ms, Cohen's $d = 1.33$, 95% CI [0.98, 1.68]) and random DTTs ($t(59) = -12.2$, $p_{adj} < 0.0001$, 95% CI of difference in RT: [$-23.4$, $-16.8$] ms, Cohen's $d = 1.57$, 95% CI [1.19, 1.95]). Finally, reaction times in incongruent DTTs were significantly slower than in random DTTs ($t(59) = -3.28$, $p_{adj} = 0.005$, 95% CI of difference in RT: [$-5.51$, $-1.34$] ms, Cohen's $d = 0.42$, 95% CI [0.16, 0.69]).

**Modelling results**

**Hierarchical generative modelling of behaviour.** When using standard linear inference statistical techniques as above, the analysis relies on averages of observed choices. For example, above we interpreted the difference of optimal responding between congruent and incongruent context as evidence that the learned habit has an influence on optimal responding (Fig. 3C). The question then arises whether we can learn more about the underlying interaction between habitual and goal-directed controllers by more careful and fine-grained computational modelling. We used computational modelling to test the assumption that the influence of the habitual action varies throughout the experiment and depending on the type of DTT, since habits strengthen through repetition. Furthermore, since here the habit is an action sequence, the effect of

the habit in a DTT should depend on the actions performed leading to that trial, and how often the corresponding sequence of actions was performed in the past[25]. Lastly, since the repeating action sequence is sometimes congruent and sometimes incongruent with the participants' goals of responding optimally, we hypothesized that this might lead to different impacts of the habit depending on the context (i.e., congruent or incongruent).

We used two concrete ways to model varying habit influence. First, we allowed the strength of the learned habitual action sequence to grow over the course of the experiment by implementing a repetition-bias $R(a)$ whose value for a given action sequence increases the more often the sequence was executed (see iii). Second, we expected participants to try and maximize their performance in a context-specific fashion. We assumed that participants use the habitual action sequence in STTs and congruent DTTs to increase performance, but suppress the influence of the habit in incongruent DTTs. When analysing average optimal responding in DTTs at group-level (Fig. 3C), we did not see such a suppression of habitual responding. However, this might have been due to averaging effects. More precise computational modelling of choice behaviour allows us to test this hypothesis at a more fine-grained participant level while taking advantage of inter-trial variations of responses, and circumventing the ceiling effect of optimal responding around 100% by using a softmax choice rule. We implemented these two aspects (incremental increase of habitual controller strength, and modulation of habitual responding depending on context) with six different models in two model families.

Three different models in Model Family 1 computed a repetition-bias based on the action sequences a participant has actually performed in the past, while Model Family 2 only models mean goal-directedness for individual DTT types. We find that models that compute repetition-biases outperform those without a repetition-bias (average WAIC 43 330 vs. 44 306, WAIC: widely applicable information criterion) (Fig. 5). We further find that the two best-performing models allow for different effect strength of the habit in the congruent and the incongruent context (models 2 (WAIC 43 351 ± 290) and 3 (43 196 ± 290) vs model 1 (WAIC 43 443 ± 289)). However, this effect is weak since the WAICs of models 1 through 3 are all within two standard errors from each other. We therefore also computed approximate Bayes factors for individual participants based on the difference in Expected Log Predictive Density ($\Delta$ELPD) derived from the WAIC. This individual-level Bayes factor analysis corroborated the findings that models 2 and 3 outperform model 1. Specifically, we found substantial evidence (Bayes factor > 3.0) in favour of model 3 over model 1 for 30 out of 60 participants (as opposed to 7 participants for model 1), and in favour of model 2 over model 1 for 20 out of 60 participants (as opposed to 4 participants for model 1).

As shown in our posterior predictive plots (Supplementary Fig. S6), model 3 reproduces the ordering and magnitude of optimal-choice proportions across DTT contexts, indicating that the fitted parameters capture the context-dependent modulation of goal-directed choice. Parameter recovery analyses further indicated reliable recovery of all model parameters (see Supplementary Fig. S7).

We further explored whether participants for whom we found evidence that they adapt the use of the habit in a context-specific fashion might be able to do so either because they have more cognitive resources at their disposal, or because they are more motivated to perform well in the task. In the first case, we would expect to see an effect of age, that is, participants for whom model 3 outperforms model 1 in the model comparison should be of lower age than the other participants. In both cases, we would expect to see more optimal responding in random DTTs in participants for whom model 3 outperforms model 1. Interestingly, we see no effect of age ($t_{58} = 0.84$, $p = 0.476$, 95% CI of difference in age: $[-3.04, 6.44]$, Cohen's $d = 0.19$, 95% CI $[-0, 32, 0.69]$), and even an opposite effect of optimal responding, that is, participants for whom model 3 outperforms model 1

show lower $\theta_Q$ on day 2 ($t_{58} = -3.26$, $p = 0.002$, 95% CI of difference in $\theta_Q$: $[-1.81, -0.43]$, Cohen's $d = 0.84$, 95% CI $[0.31, 1.37]$).

**Drift-diffusion modelling.** Model comparison showed that roughly half of participants modulate the influence of the habitual action sequence depending on the task context. This modulation could be achieved by two means: Either participants increase the habit's influence before entering a congruent DTT (proactive control), or conversely, that they decrease the habit's influence within an incongruent DTT (reactive control). Proactive control is an option since the habitual action sequence allows participants to anticipate whether the upcoming habitual response will be in a high reward-probability or a low reward-probability position. The second possibility for modulating the habit's influence depending on the task context is entering each trial unprepared and then deciding upon a choice depending on the context (reactive control). Since the drift-diffusion model allows for the breakdown of a choice process into prior response tendency and evidence accumulation (an umbrella term for all cognitive processes involved in choosing an action), we can test whether participants modulate the influence of the habit before entering a DTT (changes in prior response tendency; proactive control), or whether this modulation occurs once a trial was started (changes in evidence accumulation; reactive control). We fitted a DDM choice-rule in a DDM with collapsing bounds to account for time pressure (<600 ms). Since there is no repeating action sequence in the random condition, we tested the difference in starting point bias and drift rate in both congruent and incongruent DTTs compared to random DTTs (Fig. 6). We present the analysis of the DDM on day 2 in order to avoid the influence of learning effects at the beginning of the task on day 1 (results for day 1 are however qualitatively similar, see supplementary material). We find that the drift-rate in congruent DTTs is barely significantly increased compared to random DTTs ($\mu(\Delta v) = 0.12 \pm 0.48$, $t(59) = 2.00$, $p = 0.0497$, 95% CI $[0.0002, 0.25]$, Cohen's $d = 0.26$, 95% CI $[0.00, 0.51]$), while it is significantly decreased with a large effect size in incongruent DTTs ($\mu(\Delta v) = -0.39 \pm 0.42$, $t(59) = -7.20$, $p < 0.0001$, 95% CI $[-0.50, -0.28]$, Cohen's $d = 0.93$, 95% CI $[0.62, 0.93]$). A direct comparison confirmed that the drift rate was significantly different between congruent and incongruent DTTs ($t(59) = 5.59$, $p < 0.0001$, 95% CI $[0.33, 0.70]$, Cohen's $d = 0.72$, 95% CI $[0.44, 1.00]$). The effect on the starting point bias is reversed: whereas it is significantly increased for congruent DTTs ($\mu(\Delta z) = 0.07 \pm 0.05$, $t(59) = 9.62$, $p < 0.0001$, 95% CI $[0.05, 0.08]$, Cohen's $d = 1.24$, 95% CI $[0.90, 1.58]$), the change for incongruent DTTs is not significant ($\mu(\Delta z) = 0.009 \pm 0.04$, $t(59) = 1.74$,

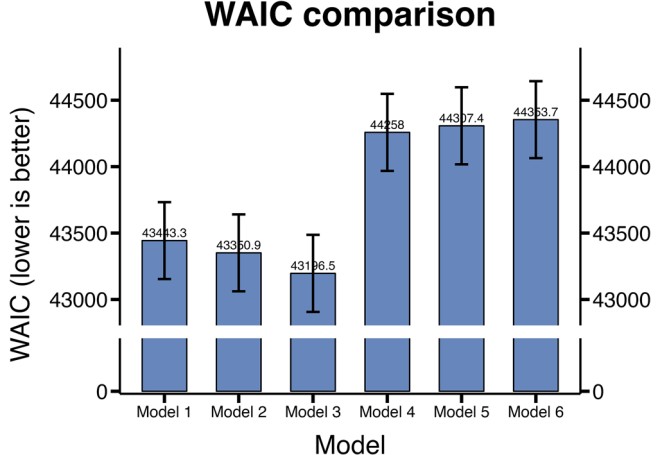

## WAIC comparison

**Fig. 5 | Model comparison.** WAIC across both days ($n = 60$). Models with repetition-bias (model 1 - model 3) outperform the models without repetition-bias (model 4 - model 6). Within models model 1 - model 3 error bars denote standard errors for WAIC.

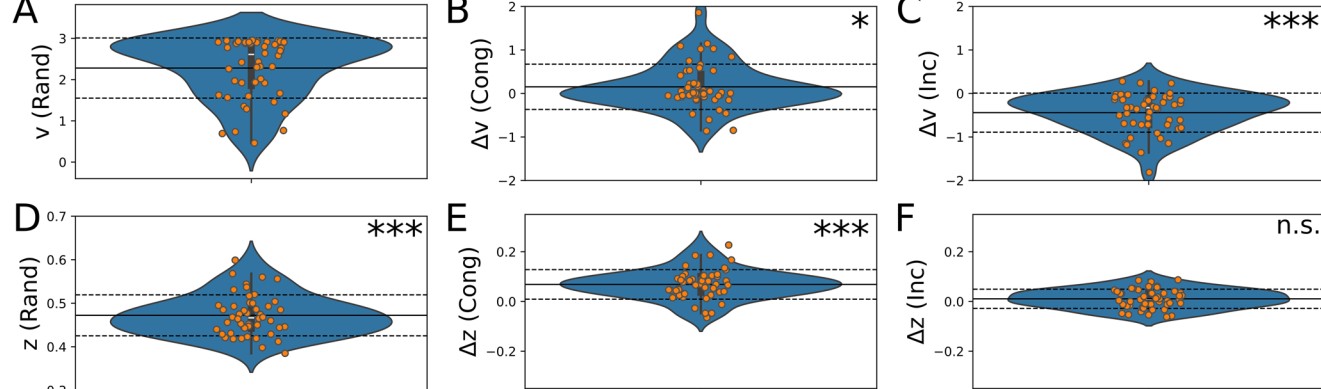

**Fig. 6 | DDM results of day 2 of the experiment ($n = 60$).** Distribution of posterior means. **A** $v(Rand)$, Drift Rate in random DTT; **B** $\Delta v(Cong)$ & **C** $\Delta v(Inc)$: Difference of drift rate in congruent and incongruent DTTs compared to random DTTs. **D** $z(Rand)$, Starting point bias in random DTTs; **E** $\Delta z(Cong)$ & **F** $\Delta z(Inc)$: Difference of starting point bias in congruent and incongruent DTTs compared to random DTTs. Orange dots denote posterior means for individual participants. A starting point bias of $z(Rand) = 0.5$ means no preference for one of the two responses, with $z(Rand) > 0.5$ indicating a preference for the optimal response option. Here, $z(Rand)$ is slightly shifted towards the suboptimal response option. Solid (dashed) lines denote group means (standard deviations). ***$p < 0.001$, **$p < 0.01$, *$p < 0.05$. $p$-values derived from one-sample $t$-tests against 0.5 ($z$ (Rand)) or 0 ($\Delta v$/ $\Delta z$).

$p$ = 0.088, 95% CI [ − 0.001, 0.018], Cohen's $d$ = 0.22, 95% CI [ − 0.03, 0.48]), compared to random DTTs. Crucially, a direct statistical comparison between congruent and incongruent contexts confirmed that the starting point shift was significantly larger in congruent compared to incongruent DTTs ($t$(59) = 6.87, $p$ < 0.0001, 95% CI [0.04, 0.08], Cohen's $d$ = 0.89, 95% CI [0.58, 1.18]). These results suggest that congruent and incongruent DTTs are treated differently by participants, again supporting our previous findings that participants adapt their response strategy depending on the DTT context. Since the habitual action sequence can be considered a prior response bias, the results further support the above hypothesis that participants exploit the habitual action sequence in congruent DTTs (starting point bias shifts towards optimal response option). This would imply that participants have learned to exploit their habit proactively in those DTTs where the upcoming sequence element is in a high-reward location. Conversely, in incongruent DTTs, they engage an additional cognitive process (e.g., for interference control).

## Discussion

In this study, we investigated the interaction between a habitual action sequence and goal-directed behaviour, using a task where habitual and goal-directed actions are either congruent or incongruent with each other. Crucially, we imposed a tight response time limit of 600ms for responding. We showed that computational models assuming a varying repetition-based habit strength outperform models that do not use this assumption. This result indicates that the influence of the learned habit is not constant throughout the experiment but varies depending on the number of performed sequence repetitions. Computational modelling further showed substantial evidence that at least 50% of participants (according to model 3) adaptively modulate the influence of the habit on action selection depending on the task context, that is, whether the habit is congruent or incongruent with the goal in dual-target trials (DTT) trials. Further, using drift-diffusion modelling, we found that the habit differentially influences prior response tendencies and evidence accumulation (information processing), depending on the context.

### Habitual action sequences strengthen through repetition

A well-established computational model of habitual behaviour models goal-directed behaviour as model-based reinforcement learning and habitual behaviour as model-free reinforcement learning[10,11]. This approach assumes that habits are reward-based, and disregards the possibility that habits might be strengthened by repetition alone[13,25]. Further, Thorndike's *law of exercise* proposes that the mere execution of an action increases its probability of being chosen again in the future[41]. Here, using computational modelling, we find evidence that habits are learned through repetition only, and that the habit strength varies with the number of repetitions performed in the past. In the model, we explicitly described the habit as a sequence. This is in line with the formulation of habits as action sequences[23,25], and with experimental evidence that the basal ganglia, which are heavily involved in habitual behaviour, chunk individual actions into sequences[42]. For example[23], proposed a theoretical framework for habits as action sequences, where individual actions are chunked into one habitual action sequence. We here found that learning the action sequence led to both reduced error rates and shorter reaction times. Importantly, these reaction-time benefits also depended on task context. When the habitual sequence was congruent with the goal-directed response, sequence execution facilitated performance, resulted in faster responses and shifted the bias (DDM starting point) towards the high. By contrast, when the habitual response conflicted with the goal-directed action, reaction times were slowed, consistent with the need to inhibit a prepotent, repetition-driven response and mainly effected the evidence-accumulation process. Together, these findings link the observed reaction-time effects directly to the underlying mechanism of repetition-based sequence learning and support characterisations of

habits as fast and automatic, yet relatively inflexible processes that can interfere with goal-directed behaviour when task demands or outcomes change.

Although we did not collect neural data in the present study, our findings are in principle compatible with evidence for overlapping and interacting corticostriatal circuits[17] as well as arbitration and control mechanisms[16,18,19] that flexibly adjust the influence of more deliberated goal-directed or more simple stimulus-response control over behavioural output. Future research will have to elucidate how reinforcement and sequence repetition combine to create habitual action sequences like the one observed here.

### Habits are modulated depending on context

Model comparison showed that according to the best model 3, for approximately 60% of participants the influence of a habit is modulated by context in DTTs, that is, the modulation depends on whether the habit is congruent or incongruent with the participants' instructed goals. For those participants who show evidence of such a modulation, habit influence in incongruent DTTs is reduced (slower RTs, compared to congruent DTTs. As congruent and incongruent DTTs are rare (15% of all trials) and are interspersed with STTs, this means that the habit influence is modulated adaptively in a trial-specific fashion. It is likely that one dominant factor for limited context-specific responding was the rather demanding response time limit of 600ms. Using drift-diffusion modelling, we further investigated mechanisms that may explain the trial-wise modulation of habit influence. We found a context-specific influence of the learned action sequence on choices. In congruent trials, the congruency between goal and habit increases proactive control, which manifests as a shift of the prior response tendency towards the optimal response. The fact that no prior response tendency towards the optimal response is observed in random DTTs suggests that goal-directed processes only create a prior response tendency when interacting with the habitual response, but are not used to create a prior response tendency on their own. In incongruent trials, the incongruency between goal and habit appears to lead to a reactive control process, which is evidenced by a decreased drift rate and higher RTs. This might be the result of an adaptive inhibition of the habitual response, in favour of increased optimal responding. Such suppression is well known in the field of executive function research as interference control[43]. As outlined in the introduction, contextual inference can be computationally appealing because it enables fast and flexible adaptation. This perspective builds on work in motor control[44,45], learning and memory retrieval[46], and decision-making under uncertainty[25,26], and fits well with our paradigm, in which context detection is presumably comparatively cheap. We emphasize that this claim is task-dependent and does not necessarily generalize to all real-world settings. When contexts are few, stable, and/or signalled by reliable cues, inferring contexts can reduce policy complexity and expected computational costs by allowing the reuse of context-specific action mappings. In contrast, when contexts are numerous, rapidly changing, or weakly signalled, context inference itself may become costly and outweigh these benefits. From a resource-rational perspective, the computational efficiency of contextual inference therefore depends on environmental structure, learning history and expected outcomes.

### Relation to dual-process and conflict monitoring accounts

A related perspective comes from the dual-process literature on conflict detection under time pressure, particularly the work by De Neys and colleagues. Here the idea is that when responses are generated rapidly and intuitively, conflicting response tendencies are often detected automatically, leading to measurable slowing and performance costs, without invoking slow, reflective deliberation[47–49]. Crucially, such conflict detection has been shown to operate under severe time constraints comparable to those in our task. From this perspective, the congruency effects observed does not necessarily imply deliberative "System 2" intervention. Instead, they are

compatible with a fast, automatic conflict-monitoring mechanism that detects misalignment between a prepotent response tendency (here, the habitual action sequence) and current task goals (reward maximization). In our computational framework, this conflict detection is expressed as a context-dependent modulation of the influence of repetition-based habit values on choice, rather than as a separate deliberative controller. Thus, from one perspective, the habit-goal interaction studied here can be understood as a computationally explicit instantiation of the fast conflict-detection processes emphasized in contemporary dual-process accounts. On the other hand, the work of De Neys and colleagues has primarily focused on belief- and judgment-based decisions e.g., refs. 47,50 rather than habit-like motor action sequences or reward maximization. Accordingly, further research is warranted to clarify the overlap and/or domain-specific differences between both lines of work.

### Limitations

A few limitations need to be addressed. First, learning of the sequence and task context is not independent of high reward locations that stayed constant over the course of the task. Therefore, entering a high reward location influenced behaviour in STT and DTT trials (see Supplementary Information). Importantly, while these reward location effects do exist, our supplementary analyses confirm that RT effects in dual-target trials persisted after controlling for reward location. This suggests that participants do not merely respond faster because a habitual action leads to a location associated with higher reward but also adapt the influence of the habitual sequence depending on whether it aligns with that goal or not. It is important to note that the habitual sequence studied here develops over minutes to hours rather than weeks or months. While real-world habits such as making coffee or smoking arise from hundreds of repetitions across diverse contexts, the present paradigm aims to capture a core mechanistic component of habit formation, namely repetition-driven automatization, on an experimentally tractable time scale. Similar short-term sequence-learning paradigms have long been used to study procedural habits[6,27,28,30]. Thus, our findings should not be interpreted as modelling long-term behavioural routines but rather as revealing how rapidly acquired habit-like action sequences can be adaptively modulated by context under time pressure. We acknowledge that uncertainty remains regarding the extent to which this paradigm generalizes to real-world habits and that studies with longer time scales (e.g., with gamified smartphone task) should be conducted. Finally, we did not collect additional measures to assess whether other factors, i.e., motivational factors, attentional mechanism influenced task performance. Further studies are necessary to investigate what factors modulate decision weights.

In summary, the present findings show that approximately 50% of participants modulate their learned habit in a context-specific fashion in order to increase optimal responding under time pressure. This rapid and adaptive modulation allows them to use their habit in a goal-directed manner when it is beneficial to performance, and minimize its impact on performance when acting habitually would be detrimental. In the present case, the context is defined by the congruency between habitual and goal-directed action, which raises the question of the nature of interaction between goal-directed and habitual control. Further research is warranted to investigate whether such context-specific interaction underlies the balancing between habitual and goal-directed control in general.

Given that only around 60% of participants were able to adaptively allocate inhibitory control over the habit in the incongruent context, future research should investigate whether such individual differences are related to mental disorders of maladaptive habits, such as addiction.

### Data availability

The experimental paradigm with which data collection was done (in javascript), raw data and curated CSV files are available on the Open Science Framework: https://osf.io/67txz/?view_only=5dec53c3a89043fc954ce8d512d7bcfc.

### Code availability

Analysis code for behavioural analyses and computational modelling in Python and R is available on the Open Science Framework: https://osf.io/67txz/?view_only=5dec53c3a89043fc954ce8d512d7bcfc.

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

## Acknowledgements

Funded by the German Research Foundation (DFG, Deutsche Forschungsgemeinschaft), SFB 940 - Project number 178833530, TRR 265 - Project number 402170461, GRK 2773 - Project number 454245598, and as part of Germany's Excellence Strategy - EXC 2050/1 - Project number 390696704 - Cluster of Excellence "Centre for Tactile Internet with Human-in-the-Loop" (CeTI) of Technische Universität Dresden. B.J.W. position at University Hospital Tuebingen is funded by an Alexander von Humboldt Professorship awarded to Peter Dayan (Max-Planck-Institute for Biological Cybernetics). The funders had no role in study design, data collection and analysis, decision to publish or preparation of the manuscript.

## Author contributions

S.F., M.S. and S.K. conceptualised the experiment. S.F., B.J.W., S.S., M.S. and S.K. conceptualized data analysis. S.F. collected the data. B.J.W. and S.F. analysed the data. B.J.W. conceptualized data analysis with drift-diffusion model. S.F. and B.J.W. wrote the first version of the manuscript. All authors interpreted the results and revised the manuscript. M.S. and S.K. supervised the project.

## Funding

## Competing interests

The authors declare no competing interests.
