## [Transparent Peer Review file · Communications Psychology]

Context-Dependent Interaction Between Goal-Directed and Habitual Control Under Time Pressure

Corresponding Author: Dr Ben Wagner

Version 0:

Decision Letter:

Dear Mr Frölich,

Thank you for your patience during the peer-review process. Your manuscript titled "Context-Dependent Interaction Between Goal-Directed and Habitual Control Under Time Pressure" has now been seen by 3 reviewers, whose comments are appended below. You will see that they find your work of some potential interest. However, they have raised quite substantial concerns that must be addressed. In light of these comments, we cannot accept the manuscript for publication, but would be interested in considering a revised version that fully addresses these serious concerns.

We hope you will find the Reviewers' comments useful as you decide how to proceed. Should additional work allow you to address these criticisms, we would be happy to look at a substantially revised manuscript. If you choose to take up this option, please highlight all changes in the manuscript text file, and provide a detailed point-by-point reply to the reviewers.

Editorially, we consider it crucial that the reviewers' concerns about the paradigm, the sample, and the modelling evidence are comprehensively addressed, through substantive additional data analysis and potentially new empirical data.

Please ensure you follow our statistical guidelines when reporting statistics (<https://www.nature.com/commspsychol/submit/submission-guidelines#statistical-guidelines>). Where you interpret null results, this interpretation must be based on Bayes Factors, equivalence tests, or suitable quantitative model comparison.

I am attaching a checklist that details critical reporting requirements for the revised manuscript. Please attend to each item and ensure your manuscript is fully compliant. We are requesting that your manuscript aligns with these requirements as this facilitates the evaluation of your manuscript, reducing delays in re-review and potential future acceptance. If your revised manuscript is not aligned with these requests on major issues, such as those concerning statistics, it may be returned to you for further revisions without re-review. Additional information can be found in our style and formatting guide Communications Psychology formatting guide.

If the revision process takes significantly longer than five months, we will be happy to reconsider your paper at a later date, provided it still presents a significant contribution to the literature at that stage.

Please use the following link to submit your
- revised manuscript,
- point-by-point response to the referees' comments,

- cover letter (as a separate document),
- the Editorial Policy Checklist (see below),
- the Reporting Summary (see below), and
- the completed Editorial Request Table (attached):

Link Redacted

Thank you for the opportunity to review your work.

Best regards,

Troby Lui

Troby Lui, PhD
Associate Editor
Communications Psychology

REVIEWER EXPERTISE:

Reviewer #1: habit/goal-directed motor behaviour, context in decision-making

Reviewer #2: cognitive control, computational modelling

Reviewer #3: habit, context, computational modelling

REVIEWER REPORTS:

Reviewer #1 (Remarks to the Author):

The present manuscript describes one experiment in which participants were assessed for performance on a task meant to measure flexibility in habitual responding gated by 'response context' (i.e., whether or not the previous response was congruent or incongruent with the target habitual response). The authors find that performance for individual items learned as part of a sequence are faster than if they were randomized. Further, they find that performance of a goal-directed behavior is better when they align with the learned sequence. They then use several computational models to determine which best describes the current data and suggest that repetition, rather than reinforcement, is what contributes to development of a habit.

The paper is well-written and I think makes some strides in connecting rodent findings to human work, but ultimately I think the connection to habitual performance could be made more clear.

- 1) A traditional view of habitual behavior is if it persists despite the reinforcer losing value in some way. How did the present experiment determine that behavior was indeed habitual? This is unclear.
- 2) One of the difficulties in interpreting the present results is that they introduce a notion of response conflict. It seems as though responses that are not in conflict with what is expected are faster than those that are. I think some further explanation of how this is defined as habit would be especially beneficial here.
- 3) The authors state that 60 of the eligible 65 participants were analyzed in the present study. How were the 5 participants chosen for exclusion? This seems arbitrary.
- 4) Finally, I'm not sure how the authors disentangle response sequence (which I believe is their targeted habitual behavior) from context here (which is supposed to be modulating behavior).

Reviewer #2 (Remarks to the Author):

Frölich et al present data acquired in the context of an online study addressing the timely question of how repetition of action sequences (i.e. habits) and goal-directed actions interact to facilitate responding in congruent or harm responding in incongruent contexts. For inference they extend their analysis by computational models that allow teasing apart distinct mechanisms of dynamic habit strength adaptation and proactive versus reactive control. They show some form of context-dependent adaptations based on habits, which seem to be more prevalent in the congruent domain. While I judge the overall quality and clarity of the manuscript very positively, I would invite the authors to clarify several methodological issues (e.g. lack of model validation, confounds) and enhance the embedding of findings in the broader literature.

Major:

- 1) It is impossible for readers to evaluate whether the winning model is a suitable model and able to successfully reproduce the relevant behavioural findings. Thus, I encourage the authors to conduct post-hoc model validation as proposed in e.g. Wilson & Collins (2019).

2) Authors interpret their results as a context effect based on the context manipulation (i.e. congruency vs incongruency between habit and goal-directed action). If I understand the task correctly, the conditions are confounded by the acquired action sequence entering a high- (i.e. in congruent) vs low-reward (i.e. in incongruent) action and I would need additional analyses to be convinced that the effect is not driven by this confounding explanation. First, does the presence of high- vs low-reward locations affect task performance in SSTs, such as invigoration of RTs or adaptation of bias as a function of (the expectation of) reward? And secondly, do congruency effects in response times, initial bias and drift rate effects go beyond (expected) reward effects in STTs where no interference should be present? I wonder whether the matching STTs would be a suitable reference on top of the currently contrasted DTTs from the random condition.

3) At this point, it is unclear which hypotheses are tested by which kind of analyses and whether results align with expectations. Towards the end of the results section a subgroup of participants is defined based on their successful adaptation to context. For now, it is unclear whether this appears post-hoc to be an interesting group or whether this is the group of participants that matches a predefined hypothesis. A clear distinction between a priori expectations and follow-up exploratory analyses, which could be more elaborate, would be helpful. If this group happens to be an interesting group, it would be interesting to show behavioural data split for the two groups in an exploratory fashion.

4) It is important to acknowledge that the sample size is surprisingly small, considering 4 counterbalancing schemes of 15-18 participants and I was surprised that identical number of participants in each group was prioritized above power. Do results hold when 65 participants are included? If counterbalancing group is a relevant factor, it should be shown that results align, or control analyses should be included.

Minor:

Authors talk about dynamic development of habit but do not present any time-resolved evidence for this point in raw data or models but in fact focus on second day. Focusing on the second day makes sense given the research question, but evidence of the development of a habit should be presented.

It was a challenge to understand the task (instructions) and I am still not sure I do after looking into the source paper. I would encourage the authors to put more information on the task in the main text and illustrate multiple trials in the task schema. Additionally, it would be helpful to address the remaining questions: Are reward positions relevant for the acquisition of the action sequence? What kind of learning process is expected to be captured by temporal difference learning if responses are explicitly cued by target? Are the high-reward locations constant throughout one day?

Consider moving Fig 2 to supplements (common DDM approach).

Fig 3: Please include individual data points and connect points for within-subject data. Bar chart carries very little information.

Why do authors follow up Bayesian models using frequentist statistics instead of presenting group-level parameters in main text and e.g. highest density intervals for effects of interest. I am missing a figure on group-level parameters of winning model and interpretation of parameters.

Regarding DDMs: I wondered whether i) the context effects on drift rate and starting point are related, ii) there is a significant starting point bias in congruent-only and iii) a contrast between incongruent vs congruent DTTs results in the same conclusions while perhaps giving more specific answers about use of information on top of habit?

Did authors consider probing the congruency by offering no target instead of DTTs? What would be expected?

Embedding of findings in literature (mainly discussion) should be more elaborate.

Suppl results figure S1: where do top and bottom refer to, what are the colours?

Suppl results figure S2: do left and right represent day 1 and 2?

Correct sentence? p.2 "This would mean that the goal-directed controller always performs its computations which makes the speed of habit execution, relative to goal-directed control, would be hard to explain"

Reviewer #3 (Remarks to the Author):

Review for Frölich et al.

The authors investigate the computational processes underlying the arbitration between habits and goal-directed actions, providing evidence that habits are learned through pure repetition and that arbitration is flexible, depending on the context.

The paper is well-written and relevant to a broad readership. The hypotheses are clearly stated. Overall, I enjoyed the paper, but I do have several remarks and points that I would like the authors to address.

First, regarding the modeling procedure, I found the paper below the standards of the field. Unless I am mistaken, several critical steps for validating the model-based results seem to be missing. Specifically, I could not find clear demonstrations of model and parameter recovery, as well as simulation analyses across models to determine which features of the data are not captured by the losing model. Such analyses are crucial to link model comparison results to specific observable behaviors (see Palminteri et al., 2017, and Wilson & Collins, 2019).

Second, I am concerned that the authors do not fully acknowledge the complexity of studying habits over such short time frames. Any real-world definition of habit involves behavioral processes acquired over days, months, or even years—not within the short temporal scale investigated in this task. This issue also applies to papers cited in the manuscript, such as Wood et al. (2002). At a minimum, the authors should discuss the limitations of extrapolating behavioral propensities acquired within minutes in a lab task to real-world habitual behaviors such as making coffee, brushing teeth, or smoking—behaviors that emerge after thousands of trials over long periods.

Third, related to the previous point, the authors report findings that align with the hypothesis of distinct neural systems (dorsal vs. ventral striatum) for habits and goal-directed actions. However, what is the actual evidence for this dissociation in humans? I could not find convincing, recent evidence supporting this claim among the citations provided. I do not believe that these neural hypotheses can be considered well-supported or consensual. The authors should clarify the empirical basis for this assumption.

Fourth, the authors propose that arbitration via context detection is computationally less costly than the “traditional” method. However, this claim depends on a clear estimate of the cost of context detection itself. In the current task, context detection is trivial and effectively built into the model as an observable variable. But what about real-life situations? Is context detection always easier? This remains unclear to me. I would like to see the authors provide more justification and discussion on this point.

Fifth and finally, I believe the authors are overlooking an important body of literature on dual-controller systems, particularly regarding the effects of time pressure on performance. In this respect, they should consider the work of Wim De Neys and his reviews. In my view, their task is strongly connected to the system 1/system 2 framework, which seems more plausible than the concept of habit formation given the short time frame of their study.

EDITORIAL POLICIES

We ask that you ensure your manuscript complies with our editorial policies and reporting requirements.

To that end, we require revised manuscripts to be accompanied by two completed items: a reporting summary that collects information on study design and procedure, and an editorial policy checklist that verifies compliance with all required editorial policies

- <https://www.nature.com/documents/nr-reporting-summary.zip>>Nature Research Reporting Summary
- <https://www.nature.com/documents/nr-editorial-policy-checklist.pdf>>Editorial Policy Checklist

All points on the policy checklist must be addressed. Your revised manuscript can only be sent back to the referees if these checklists are completed and uploaded with the revision.

Notes: If you have submitted a Stage 1 Registered Report, Review, Primer, Comment, or Perspective you do not need to submit these forms. If you have already submitted these forms, you may disregard this request.

Communications Psychology is committed to improving transparency in authorship. As part of our efforts in this direction, we are now requesting that all authors identified as 'corresponding author' create and link their Open Researcher and

Contributor Identifier (ORCID) with their account on the Manuscript Tracking System prior to acceptance. ORCID helps the scientific community achieve unambiguous attribution of all scholarly contributions. You can create and link your ORCID from the home page of the Manuscript Tracking System by clicking on 'Modify my Springer Nature account' and following the instructions in the link below. Please also inform all co-authors that they can add their ORCIDs to their accounts and that they must do so prior to acceptance.

Version 1:

Decision Letter:

Dear Dr Wagner,

Your manuscript titled "Context-Dependent Interaction Between Goal-Directed and Habitual Control Under Time Pressure" has now been seen by our reviewers, whose comments appear below. In light of their advice I am delighted to say that we are happy, in principle, to publish a suitably revised version in Communications Psychology.

We therefore invite you to revise your paper one last time to address the remaining concerns of our reviewers and a list of editorial requests. At the same time we ask that you edit your manuscript to comply with our format requirements and to maximise the accessibility and therefore the impact of your work.

EDITORIAL REQUESTS:

SUBMISSION INFORMATION:

OPEN ACCESS:

*** TRANSPARENT PEER REVIEW:** Communications Psychology uses a transparent peer review system. On author request, confidential information and data can be removed from the published reviewer reports and rebuttal letters prior to publication. If you are concerned about the release of confidential data, please let us know specifically what information you would like to have removed. Please note that we cannot incorporate redactions for any other reasons.

*** CODE AVAILABILITY:** All Communications Psychology manuscripts must include a section titled "Code Availability" at the end of the methods section. We require that the custom analysis code supporting your conclusions is made available in a publicly accessible repository at this stage; please choose a repository that generates a digital object identifier (DOI) for the code; the link to the repository and the DOI must be included in the Code Availability statement. Publication as Supplementary Information will not suffice.

* DATA AVAILABILITY:

Link Redacted

Best regards,

Troy Lui

Troy Lui, PhD
Associate Editor
Communications Psychology

REVIEWERS' COMMENTS:

Reviewer #1 (Remarks to the Author):

The authors have addressed most of my concerns but one remains as the authors failed to address this in the revision: The authors state that 60 of the eligible 65 participants were analyzed in the present study. How were the 5 participants chosen for exclusion? This seems arbitrary. How exclusions were made should be fully explained in the text.

Reviewer #2 (Remarks to the Author):

The authors have done an excellent job revising the manuscript.

In particular, the addition of model validation, parameter recovery, and control analyses addressing potential confounding effects has substantially strengthened the work. These revisions have improved the data analyses, enhanced the clarity of the figures, and overall increased the quality of the manuscript.

While I am generally convinced that the manuscript presents well-supported findings, it remains unclear why reported results are based on the random exclusion of five participants rather than presenting the analyses for the complete sample. In their responses to Reviewers 1 and 2, the authors state that "the qualitative results, including the model ranking, remain unchanged." Given this clarification, I recommend that the full-sample analyses be included in the Supplementary Materials to ensure transparency and to allow readers to evaluate the results based on the complete dataset.

Reviewer #3 (Remarks to the Author):

The authors did a satisfactory work. There are some typos in the citations, I believe (Neys & Wim ?)

Response to the Editor and Reviewers for Wagner & Frölich et al., “Context-Dependent Interaction Between Goal-Directed and Habitual Control Under Time Pressure” in Communications Psychology

Manuscript ID: COMMSPSYCHOL-25-0008-T

General remarks:

We thank the Editor and the three Reviewers for their evaluation of our manuscript and for their helpful and constructive feedback. We have carefully responded to all comments and incorporated revisions, as detailed below. Furthermore, we apologize for the delay in resubmitting our manuscript. During this period, the first author transitioned to an industry position, which led to a reorganization of responsibilities within the author team. As a result, we adjusted the lead authorship and added a co-author to reflect the current contributions to the project. These changes contributed to the extended revision timeline. We have now dedicated effort to ensure a thorough and thoughtful revision.

Reviewer #1 (Remarks to the Author):

The present manuscript describes one experiment in which participants were assessed for performance on a task meant to measure flexibility in habitual responding gated by ‘response context’ (i.e., whether or not the previous response was congruent or incongruent with the target habitual response). The authors find that performance for individual items learned as part of a sequence are faster than if they were randomized. Further, they find that performance of a goal-directed behavior is better when they align with the learned sequence. They then use several computational models to determine which best describes the current data and suggest that repetition, rather than reinforcement, is what contributes to development of a habit.

The paper is well-written and I think makes some strides in connecting rodent findings to human work, but ultimately I think the connection to habitual performance could be made more clear.

Response: We thank the reviewer for their positive evaluation of our manuscript and for their helpful suggestions, we will respond to each point below.

Reviewer #1: *A traditional view of habitual behavior is if it persists despite the reinforcer losing value in some way. How did the present experiment determine that behavior was indeed habitual? This is unclear.*

Response:

We thank the reviewer for that comment. In the revision, we added a paragraph in the introduction clarifying how our operationalization of habits relates to this traditional view of habitual behavior. On **p2** we now write:

“This relates to classical accounts of habit formation in two ways. First, repeated execution of an action or a sequence of actions strengthens performance, consistent with theories that

conceptualize habits as learned action sequences \cite{dezfouli2012habits, dezfouli2014habits}. Second, the design allows us to test whether the learned sequence continues to influence behaviour when they no longer lead to high-reward outcomes, i.e. whether habitual responding persists under devaluation of the action-sequence, i.e. when the habitual and goal-directed responses conflict. In consequence we do test whether participants learn to modify the strength of the habitual action in a context-specific fashion.”

Reviewer #1: *One of the difficulties in interpreting the present results is that they introduce a notion of response conflict. It seems as though responses that are not in conflict with what is expected are faster than those that are. I think some further explanation of how this is defined as habit would be especially beneficial here.*

We thank the reviewer for this opportunity to clarify and agree that the congruent-incongruent manipulation introduces response conflict. Crucially, in our task this conflict reflects competition between a response that is favoured by extensive prior repetition under strict time pressure (the habitual action sequence) and a response that is favoured by explicit reward contingencies (the goal-directed action). The habitual component is defined not merely by expectancy, but by its repetition-based, outcome-independent strengthening and its expression as part of an action sequence. In the congruent context, the habitual component aligns with the goal-directed response, leading to faster reaction times due to the automatic execution. In the incongruent context, the same habitual tendency conflicts with the goal-directed response, producing slower reactions times as participants must inhibit the prepotent, repetition-driven response. This pattern is consistent with classical accounts of habits as fast, automatic and “relatively” inflexible responses that can interfere with goal-directed control when outcomes change.

We have expanded our discussion on p14.

“We here found that learning the action sequence led to both reduced error rates and shorter reaction times. Importantly, these reaction-time benefits also depended on task context. When the habitual sequence was congruent with the goal-directed response, sequence execution facilitated performance, resulted in faster responses and shifted the bias (DDM starting point) towards the high. By contrast, when the habitual response conflicted with the goal-directed action, reaction times were slowed, consistent with the need to inhibit a prepotent, repetition-driven response and mainly effected the evidence-accumulation process. Together, these findings link the observed reaction-time effects directly to the underlying mechanism of repetition-based sequence learning and support characterizations of habits as fast and automatic, yet relatively inflexible processes that can interfere with goal-directed behavior when task demands or outcomes change.”

Reviewer #1: *The authors state that 60 of the eligible 65 participants were analyzed in the present study. How were the 5 participants chosen for exclusion? This seems arbitrary.*

Response: We thank the reviewer for raising this point. Prior to data collection, we planned a fully counterbalanced design (differing in high reward location and block order). For completeness, we re-ran the key analyses including all 65 participants and can confirm that the qualitative results, including the model ranking, remain unchanged.

Reviewer #1: *Finally, I'm not sure how the authors disentangle response sequence (which I believe is their targeted habitual behavior) from context here (which is supposed to be modulating behavior).*

Response:

We thank the reviewer for the opportunity to further elaborate on this. While the habitual component is the action sequence acquired through repetition, the context by contrast is defined by the current alignment between the habitual response and goal-directed reward contingencies, that is, whether the habitual action happens to coincide with the optimal (congruent context) or suboptimal (incongruent context) or the absence of any relationship (random context) in goal-directed choice in the dual target trials. In other words, context is not equal to the habit itself but determines whether it is beneficial if the habitual influence is expressed or inhibited. Our modelling explicitly disentangles these two components: the repetition-values are computed purely from the history of performed sequences, while the context-modulation parameter θ_{Inc} governs how strongly these values enter the choice function.

We further elaborate on this in the methods section on p5 as follows:

“By design, the habitual and contextual components are computationally separable. The habitual component reflects a repetition-based action sequence learned from past sequence executions, whereas the contextual component is defined by the alignment between this habitual tendency and the goal-directed optimal action (high-reward location) on a given dual target trial (congruent vs. incongruent) or the absence of a sequence (random context). We compared six candidate models from two model families, one of which implements context-dependent modulation of the habitual component via a dedicated parameter. Model fitting was performed with hierarchical stochastic variational inference in pyro 1.8.6 with python 3.11.6 and within RStudio 2023.12.1 and JAGS 4.3.2 followed by formal model comparison.”

Reviewer #2 (Remarks to the Author):

Frölich et al present data acquired in the context of an online study addressing the timely question of how repetition of action sequences (i.e. habits) and goal-directed actions interact to facilitate responding in congruent or harm responding in incongruent contexts. For inference they extend their analysis by computational models that allow teasing apart distinct mechanisms of dynamic habit strength adaptation and proactive versus reactive control. They show some form of context-dependent adaptations based on habits, which seem to be more prevalent in the congruent domain. While I judge the overall quality and clarity of the manuscript very positively, I would invite the authors to clarify several methodological issues (e.g. lack of model validation, confounds) and enhance the embedding of findings in the broader literature.

Response: We thank the reviewer for their positive evaluation of our manuscript and for their helpful suggestions. We respond to each point below.

Major:

Reviewer #2: *It is impossible for readers to evaluate whether the winning model is a suitable model and able to successfully reproduce the relevant behavioural findings. Thus, I encourage the authors to conduct post-hoc model validation as proposed in e.g. Wilson & Collins (2019).*

Response: We thank the reviewer for making that point and agree that we should show whether the winning model is suitable in more detail. To do so we conducted posterior predictive checks (PPCs) to see whether the winning model reproduces the observed patterns in the percentage of optimal responses (see Supplementary Figure S6 below). These PPCs demonstrate that model 3 successfully recovers the observed empirical differences between random, congruent and incongruent task contexts. We further conducted full parameter recovery for this model across the whole range of the posterior distribution (see Supplementary Figure S7 below). Our analysis shows that all parameters show reasonable recovery.

We now describe these analyses in the methods section of the paper, refer to them in the results section and report them in the Supplementary information.

We now describe these analyses in the methods section on p8:

“Posterior predictive checks

“We performed posterior predictive checks to assess whether the best-fitting model reproduced key empirical patterns in goal-directed responding across task contexts. We therefore used posterior draws from the fitted hierarchical model to simulate binary goal-directed choices on DTTs for each participant. We then aggregated simulated choices in the same way as the observed data, computing the proportion of optimal (goal-directed) choices separately for each DTT type (random, congruent, incongruent) and directly compared them to the observed group-level proportions.”

“Parameter recovery

We performed parameter recovery for the best-fitting model (model 3). During MCMC, we generated posterior-predictive datasets by simulating DTT choice datasets conditional on joint post-warm-up parameter draws, across the whole posterior distribution while preserving the original trial structure, outcomes, and all regressors (subject-level learning-rate (lr) and weights θ_Q , θ_{Rep} , θ_{Inc} , each estimated separately for day 1 and day 2). Across this posterior-predictive pool, we randomly sampled 15 synthetic datasets and refit each dataset with the same hierarchical implementation as in the main analysis. Parameter recovery was assessed by computing Pearson correlations between the known generating parameter values (from the joint posterior draw used to simulate each dataset) and the recovered parameter estimates (posterior means), computed separately for each parameter and day and pooled across subjects and simulated datasets.”

Posterior predictive checks

Supplementary Figure S6 Group-level posterior predictive checks for the winning model 3. The violin plots and black points show the empirical percentage of optimal responses for each participant across the three DTT contexts (random, congruent and incongruent). Black horizontal bars represent the empirical group means with their SE. Red points and red connecting line depict the posterior predictive mean estimates and their SE obtained by simulating choices from the posterior distribution of model 3.

And in the results section on p11, we added this text:

“As shown in our posterior predictive plots (Supplementary Figure S6), model 3 reproduces the ordering and magnitude of optimal-choice proportions across DTT contexts, indicating that the fitted parameters capture the context-dependent modulation of goal-directed choice. Parameter recovery analyses further indicated reliable recovery of all model parameters (see Supplementary Figure S7)”

Supplementary Figure S7 Parameter recovery for model 3 across two days, with panels A-D showing day 1 and panels E-H showing day 2. Each panel corresponds to one model parameter (A and E: learning rate (lr), B and F: θ_Q , C and G: θ_{Rep} D and H: θ_{Inc}) and plots generating parameter values drawn from the joint posterior against the recovered posterior means obtained after refitting the full hierarchical model to synthetic datasets. Because we simulate directly from the posterior, the analysis accounts for the whole within and between subject variance-covariance structure. Scatter plots of true generating values (x-axis) against posterior means after refitting (y-axis) for each parameter of model 3. Points represent individual participants across 15 simulated full datasets. Parameters showed good recovery for day 1, lr : $r = 0.80$, θ_Q $r = 0.92$, θ_{Rep} : $r = 0.69$, θ_{inc} : $r = 0.53$ and day 2, lr : $r = 0.81$, θ_Q $r = 0.98$, θ_{Rep} : $r = 0.69$, θ_{inc} : $r = 0.47$.

Reviewer #2: Authors interpret their results as a context effect based on the context manipulation (i.e. congruency vs incongruency between habit and goal-directed action). If I understand the task correctly, the conditions are confounded by the acquired action sequence entering a high- (i.e. in congruent) vs low-reward (i.e. in incongruent) action and I would need additional analyses to be convinced that the effect is not driven by this confounding explanation. First, does the presence of high- vs low-reward locations affect task performance in SSTs, such as invigoration of RTs or adaptation of bias as a function of (the expectation of) reward? And secondly, do congruency effects in response times, initial bias and drift rate effects go beyond (expected) reward effects in SSTs where no interference should be present? I wonder whether the matching SSTs would be a suitable reference on top of the currently contrasted DTTs from the random condition.

We thank the reviewer for this important point. When investigating this, we indeed found that reward location (high vs. low) speeds up RTs in single-target trials (STTs). Our additional analyses reported below show that the congruency effects persist after controlling for both trialwise reward location and individual differences in reward sensitivity. We have added these analyses and corresponding figures to the Supplement of the revised manuscript and refer to them in the Discussion, specifically in the limitations section on p15:

“First, learning of the sequence and task context is not independent of high reward locations that stayed constant over the course of the task. Therefore, entering a high reward location influenced behavior in STT and DTT trials (see Supplementary Information). Importantly, while these reward location effects do exist, our supplementary analyses confirm that RT effects in dual-target trials persisted after controlling for reward location. This suggests that participants do not merely respond faster because a habitual action leads to a location associated with higher reward but also adapt the influence of the habitual sequence depending on whether it aligns with that goal or not.”

and in the Supplementary Information:

Disentangling effects of reward location and task context.

To explicitly disentangle reward-location effects from context effects (congruent vs. incongruent action sequence) on RTs we conducted a set of control analyses. Given that there was only one option in STTs all analyses were performed using linear mixed-effects models with random effects at the participant level, thereby accounting for individual differences in baseline RTs, sensitivity to reward location and task context.

First, we tested whether block (Random vs Repetition) effects could be explained by reward location alone. We therefore fit a mixed-effects model predicting STT RTs from block condition (Random vs Repetition), participant-specific reward location (High vs Low), and their interaction:

$$RT_{STT} \sim \text{blockcondition} \times \text{rewardloc} + (1 + \text{blockcondition} + \text{rewardloc} | \text{participant})$$

This analysis revealed a main effect of reward location ($\beta = +37.5$ ms, $SE = 1.96$, $t(59.98) = 19.18$, $p < .001$), with responses to low-reward locations being significantly slower than to high-reward locations. Importantly, we also observed a robust main effect of block condition, with repetition blocks producing faster RTs than random blocks even after controlling for reward location ($\beta = -12.1$ ms, $SE = 0.97$, $t(59.99) = -11.54$, $p < .001$). We additionally found a significant interaction indicating that reward-location effects were modestly amplified in repetition blocks ($\beta = +4.68$ ms, $SE = 0.70$, $t(60.11) = 6.66$, $p < .001$). Critically, responses in repetition blocks (combined congruent and incongruent contexts) were faster than in random blocks at both high- and low-reward locations, demonstrating that repetition effects cannot be reduced to reward-location alone.

We next looked at dual-target trials (DTTs) to test whether congruency effects in the repetition block could be explained by individual differences in reward location sensitivity. To this end, we computed a participant-specific reward sensitivity measure (ΔSTT) defined as the difference in

STT RTs between low- and high-reward locations, and included this measure as a between-participant covariate in a mixed-effects model predicting DTT RTs:

$$RT_{DTT} \sim dtt_{type} + \Delta STT_{rewardloc} + dtt_{type} \times \Delta STT_{rewardloc} + (1 + dtt_{type} \mid participant)$$

This analysis revealed a main effect of DTT type such that congruent trials were significantly faster than random trials ($\beta = -17.18$ ms, $SE = 4.80$, $t(60.19) = -3.58$, $p < .001$). In contrast, incongruent trials did not differ reliably from random trials in RT ($\beta = -2.14$ ms, $SE = 3.20$, $t(60.91) = -0.67$, $p = .506$). Crucially, individual reward sensitivity (ΔSTT) did not significantly predict DTT RTs ($\beta = -0.40$, $SE = 0.24$, $t(60.01) = -1.64$, $p = .107$), nor did it interact with DTT type (both interaction terms $p > .62$). Thus, participants who showed stronger reward-location invigoration in STTs did not show larger congruency effects in DTTs. Together, these results indicate that the congruency-related RT advantage in DTTs cannot be explained by individual differences in reward sensitivity measured in STTs.

Finally, we conducted an additional control analysis that accounted for trial-wise reward location in DTTs (i.e., whether the executed action entered a high- vs. low-reward location), while still controlling for participant-level reward sensitivity:

$$RT_{DTT} \sim dtt_{type} + \Delta STT_{rewardloc} + dtt_{type} \times \Delta STT_{rewardloc} + (1 + dtt_{type} \mid participant)$$

This analysis confirmed a robust trial-wise reward-location effect, with responses entering low-reward locations being significantly slower than those entering high-reward locations ($\beta = +11.62$ ms, $SE = 0.85$, $t(57,955) = 13.62$, $p < .001$). Importantly, congruency effects remained robust after controlling for both trial-wise reward location and individual reward sensitivity: congruent DTTs were substantially faster than random DTTs ($\beta = -18.77$ ms, $SE = 1.62$, $t(60.03) = -11.60$, $p < .001$). Moreover, a direct pairwise contrast between incongruent and congruent DTTs revealed slowing for incongruent relative to congruent trials (Inc – Cong: $\beta = +15.45$ ms, $SE = 1.47$, $t(61.4) = 10.47$, $p < .001$). By contrast, the participant-level reward sensitivity covariate ΔSTT did not significantly predict DTT RTs ($\beta = -0.35$ ms, $SE = 0.24$, $t(60.04) = -1.44$, $p = .16$).

Taken together, reward location does influence response times, however, repetition effects in STTs and congruency effects in DTTs remained significant after controlling for reward location at both the trial and participant levels. Given that there were no decisions associated with STTs, we cannot investigate the effect of reward location on choice. Nevertheless, we believe that these results provide evidence that the reported congruency effects cannot be explained solely by differences in expected reward and instead reflect context-dependent interference or facilitation between the acquired action sequence and goal-directed choice.

Reviewer #2: *At this point, it is unclear which hypotheses are tested by which kind of analyses and whether results align with expectations. Towards the end of the results section a subgroup of participants is defined based on their successful adaptation to context. For now, it is unclear whether this appears post-hoc to be an interesting group or whether this is the group of participants that matches a predefined hypothesis. A clear distinction between a priori expectations and follow-up exploratory analyses, which could be more elaborate, would be helpful. If this group happens to be an interesting group, it would be interesting to show behavioural data split for the two groups in an exploratory fashion.*

Response:

We agree that this needs clarification. While it was a predefined hypothesis that participants may adapt their performance based on adaption to context it was an exploratory, post hoc decision to define this subgroup and further investigate it. That is, we explored that participants for whom we found evidence that they adapt the use of the habit in a context-specific fashion (i.e. inhibit the habit when in conflict with the goal-directed choice) might be able to do so either because they have more cognitive resources at their disposal, or because they are more motivated to perform well in the task. Based on the model comparison indicating that only a subset of participants showed strong context-dependent habit modulation, we explored whether individual differences in age or task parameters (θ_Q) might explain this variation. Although this was not part of our a priori hypotheses, we thought this provides a useful first step toward understanding why some individuals flexibly modulate habitual tendencies while others do not. We now explicitly state that this was an exploratory analysis within this section on p12.

Reviewer #2: *It is important to acknowledge that the sample size is surprisingly small, considering 4 counterbalancing schemes of 15-18 participants and I was surprised that identical number of participants in each group was prioritized above power. Do results hold when 65 participants are included? If counterbalancing group is a relevant factor, it should be shown that results align, or control analyses should be included.*

Response:

We thank the reviewer for raising this point which was also mentioned by Reviewer #1. Prior to data collection, we planned a fully counterbalanced design (equal amount of participants per task version, differing in high reward location and block order). For completeness, we re-ran the key analyses including all 65 participants and can confirm that the qualitative results, including the model ranking, remain unchanged.

Minor:

Reviewer #2: *Authors talk about dynamic development of habit but do not present any time-resolved evidence for this point in raw data or models but in fact focus on second day. Focusing on the second day makes sense given the research question, but evidence of the development of a habit should be presented.*

Response: We thank the reviewer for making that point. We believe that two findings support this interpretation and now explicitly link to those in the results section. First, given how we model the data, the influence of θ_{Rep} on choice depends on the number of repetitions and therefore exerts a stronger influence as repetition values increase, i.e. in the congruent task context. Second, θ_{Rep} increases from day 1 to day 2 ($BF_{10}=29$), highlighting an increase over time (please see Supplementary Figure S8).

Reviewer #2: *It was a challenge to understand the task (instructions) and I am still not sure I do after looking into the source paper. I would encourage the authors to put more information on the task in the main text and illustrate multiple trials in the task schema. Additionally, it would be helpful to address the remaining questions: Are reward positions relevant for the acquisition of the action sequence? What kind of learning process is expected to be captured by temporal difference learning if responses are explicitly cued by target? Are the high-reward locations constant throughout one day?*

Response:

Yes, we agree that the task description could be more detailed. In the revision we have partly rewritten and expanded the task description in the Methods section (p3 and p4) and now explicitly describe several consecutive trials (STT -> STT -> DTT) in the text. This clarifies how the repeating motor sequence progresses and how congruent vs incongruent dual-target trials arise.

First, we clarify that the high-reward locations are fixed throughout the experiment, i.e., they remain constant across both days. Second, we report new control analyses in the Supplementary Information (“Disentangling effects of reward location and task context”) that quantify the influence of reward location on RTs and confirm that task context effects cannot be explained by reward location alone. Third, regarding the role of temporal-difference learning: although participants were informed that two locations tend to yield higher rewards than the remaining two, they were not told the exact reward probabilities. The TD learning component therefore captures how participants learn these reward contingencies over experience and use the resulting value estimates to guide choices in dual-target trials. We now state this explicitly in the Methods.

e.g. on p6:

“Note that although participants were informed that some locations were generally associated with higher versus lower reward likelihood, they were not told the exact reward probabilities. Temporal-difference learning therefore captures how reward contingencies are learned over time and how the resulting value estimates are used to guide choices on DTTs”

Reviewer #2: *Consider moving Fig 2 to supplements (common DDM approach).*

Response: We understand the rationale for this comment. Still, we respectfully prefer to keep Fig. 2 in the main text because it depicts the drift–diffusion model with collapsing decision boundaries, which is a nonstandard extension of the classical DDM. Unlike the conventional fixed-boundary DDM, this variant has no closed-form likelihood and requires a neural-network-based approximation to fit the data (here implemented via the HSSM toolbox). The collapsing boundaries are critical to model the strict 600 ms response deadline used in our task, as they allow the decision threshold to decrease dynamically within a trial, which we found is essential to capture behavior under time pressure.

Reviewer #2: *Please include individual data points and connect points for within-subject data. Bar chart carries very little information.*

Response:

We agree and have recreated Figure 3 using violin plots and individual data points to provide a more transparent visualization of the data (see revised Figure 3 below).

Figure 3-revised Evidence for reward contingency learning and habitual motor sequence. } A: In single-target trials (STT) of the repeating-sequence condition (Rep), participants displayed shorter reaction times than in the random (Rand) condition. B: In STTs of the Rep condition, participants performed less errors than in the Rand condition. C: Proportion of optimal responses for dual-target trial (DTT) types random, congruent and incongruent. In all three conditions, participants responded optimally above-chance levels. The significant difference between congruent and incongruent DTTs indicates that participants learned a repetition bias. D: Reaction times are different between all three types of DTTs. Error bars indicate standard errors of the mean.

Reviewer #2: Why do authors follow up Bayesian models using frequentist statistics instead of presenting group-level parameters in main text and e.g. highest density intervals for effects of interest. I am missing a figure on group-level parameters of winning model and interpretation of parameters.

Response: We now explicitly report and visualize the group-level posterior distributions of the winning model (model 3), see Supplementary Information. Specifically, we added a new figure (Supplementary Figure 8, see below) that shows (i) the posterior distributions of all group-level

hyperparameters for both days, their 95% highest density intervals (HDIs), and their posterior difference distributions (day 2 – day 1). For each difference distribution, we additionally report directional Bayes factors quantifying evidence for increases from day 1 to day 2.

Supplementary Figure 8: Posterior hyperparameter distributions. Top row: Hyperparameter distributions for day 1. A: learning rate (lr), B: θ_Q , C: θ_{Rep} , D: θ_{inc} . Middle row: Corresponding posterior distributions for day 2: E: lr, F: θ_Q , G: θ_{Rep} , H: θ_{inc} ; Bottom row: Posterior difference distributions (day 2 – day 1). Positive values indicate higher parameter values on day 2. I: Δ lr, BF_{10} (day 2 > day 1): 7.01. J: Δ θ_Q , BF_{10} : 2.55. K: Δ θ_{Rep} , BF_{10} : 29.5. L: Δ θ_{inc} , BF_{10} : 2.71.

Reviewer #2: Regarding DDMs: I wondered whether i) the context effects on drift rate and starting point are related, ii) there is a significant starting point bias in congruent-only and iii) a contrast between incongruent vs congruent DTTs results in the same conclusions while perhaps giving more specific answers about use of information on top of habit?

Response: We thank the reviewer for bringing up these points. First, we found that changes in drift rate and response bias are indeed related: individual differences in the shift from the random context to both the congruent and incongruent contexts show moderate negative correlations (congruent: $\rho = -0.36$; incongruent: $\rho = -0.46$). Second, we observed a significant starting-point (bias) shift from the random to the congruent context (see Supplementary Figure S9 below and Figure 6 in the main manuscript), whereas the corresponding bias shift from random to incongruent was not significant. In the congruent context, participants exhibited a bias toward the optimal response, which is theoretically consistent with an automatic, habit-aligned response tendency when habitual and goal-directed actions coincide (no response conflict). Third, examining

individual parameter changes revealed a qualitative dissociation between contexts. While most participants (~ 40) showed evidence for a reduction in drift rate when transitioning from the random to the incongruent context (see Supplementary Figure S9A), most participants (~ 40) did not show a substantial drift-rate change from the random to the congruent context (see x-axis in Supplementary Figure S9B). Instead, the congruent context primarily affected response bias. Together, this pattern suggests that congruent trials are characterized by proactive, bias-based facilitation of the habitual response, consistent with automatic processing. In contrast, incongruent trials primarily affect evidence accumulation, reflected in reduced drift rates, consistent with response conflict on the level of evidence accumulation. We now report this analysis in the supplement of the paper.

Supplementary Figure S9 Posterior hyperparameter distributions. A: Association between changes in drift rate (Δv) and response bias (Δz) from the random to the incongruent context. B: Association between changes in drift rate (Δv) and response bias (Δz) from the random to the congruent context.

Reviewer #2: *Did authors consider probing the congruency by offering no target instead of DTTs? What would be expected?*

Response: That is an interesting idea. We did not do that, but in theory we believe that one would expect a sequence-congruent button press. This also resonates with the DDM bias for congruent trials only. We appreciate this suggestion and will keep it in mind for future research.

Reviewer #2: *Embedding of findings in literature (mainly discussion) should be more elaborate.*

Response: Yes, we agree. We substantially expanded our discussion section and limitations and now also discuss our findings in relation to dual system accounts, real world habits and others.

Reviewer #2: *Suppl results figure S1: where do top and bottom refer to, what are the colours?*

Response: We thank the reviewer for pointing out that error and corrected the figure caption.

Reviewer #2: *Suppl results figure S2: do left and right represent day 1 and 2?*

Response: Yes, we thank the reviewer for pointing that out and adjusted the figure caption accordingly.

Reviewer #2: *Correct sentence? p.2 “This would mean that the goal-directed controller always performs its computations which makes the speed of habit execution, relative to goal-directed control, would be hard to explain”*

Response: We revised that sentence and now write:

“In a mixing model, both the habitual and goal-directed controllers are assumed to compute their outputs in parallel. This implies that the goal-directed controller is always engaged, which makes it difficult to explain why habitual actions are executed faster than goal-directed ones.”

Reviewer #3 (Remarks to the Author):

The authors investigate the computational processes underlying the arbitration between habits and goal-directed actions, providing evidence that habits are learned through pure repetition and that arbitration is flexible, depending on the context. The paper is well-written and relevant to a broad readership. The hypotheses are clearly stated. Overall, I enjoyed the paper, but I do have several remarks and points that I would like the authors to address.

Response: We thank the reviewer for their support and address each point below.

Reviewer #3:

First, regarding the modeling procedure, I found the paper below the standards of the field. Unless I am mistaken, several critical steps for validating the model-based results seem to be missing. Specifically, I could not find clear demonstrations of model and parameter recovery, as well as simulation analyses across models to determine which features of the data are not captured by the losing model. Such analyses are crucial to link model comparison results to specific observable behaviors (see Palminteri et al., 2017, and Wilson & Collins, 2019).

Response: We thank the reviewer for making that point. We conducted posterior predictive checks (PPCs) to see whether the winning model reproduces the observed patterns in percentage of optimal responses (see Supplementary Figure S6). These PPCs demonstrate that model 3 successfully recovers the observed empirical differences between random, congruent and incongruent task contexts. We further conducted full parameter recovery for this model across the whole range of the posterior distribution (see Supplementary Figure S7). Our analysis shows that all parameters show reasonable recovery. We also introduce these analyses in the methods section of the paper.

Posterior predictive checks

Supplementary Figure S6 Group-level posterior predictive checks for the winning model 3. The violin plots and black points show the empirical percentage of optimal responses for each participant across the three DTT contexts (Random, Congruent and Incongruent). Black horizontal bars represent the empirical group means with their SE. Red points and red connecting line depict the posterior predictive mean estimates and their SE obtained by simulating choices from the posterior distribution of model 3.

Supplementary Figure S7 Parameter recovery for model 3 across two days, with panels A-D showing day 1 and panels E-H showing day 2. Each panel corresponds to one model parameter (A and E: learning rate (lr), B and F: θ_Q , C and G: θ_{Rep} , D and H: θ_{Inc}) and plots generating parameter values drawn from the joint posterior against the recovered posterior means obtained after refitting the full JAGS model to synthetic datasets. Because we simulate directly from the posterior, the analysis accounts for the whole within and between subject variance-covariance structure. Scatter plots of true generating values (x-axis) against posterior means after refitting (y-axis) for each parameter of model 3. Points represent individual participants across 15 simulated full datasets. Parameters showed good recovery for day 1, lr : $r = 0.80$, θ_Q : $r = 0.92$, θ_{Rep} : $r = 0.69$, θ_{inc} : $r = 0.53$ and day 2, lr : $r = 0.81$, θ_Q : $r = 0.98$, θ_{Rep} : $r = 0.69$, θ_{inc} : $r = 0.47$.

Reviewer #3:

Second, I am concerned that the authors do not fully acknowledge the complexity of studying habits over such short time frames. Any real-world definition of habit involves behavioral processes acquired over days, months, or even years—not within the short temporal scale investigated in this task. This issue also applies to papers cited in the manuscript, such as Wood et al. (2002). At a minimum, the authors should discuss the limitations of extrapolating behavioral propensities acquired within minutes in a lab task to real-world habitual behaviors such as making coffee, brushing teeth, or smoking—behaviors that emerge after thousands of trials over long periods.

Response:

We fully agree that real-world habits such as making coffee or smoking develop over extended time frames and involve hundreds to thousands of repetitions embedded in vastly different contextual associations. Our study is not intended to reproduce such long-term processes but to isolate and characterize one likely core mechanism of habit formation: repetition-based sequence learning, under controlled laboratory conditions. Participants in our experiment performed

approximately 6720 trials (\approx 5700 single-target and 1080 dual-target trials) across two days, which is within the range of established laboratory paradigms that induce automatic motor sequences or habit-like behaviors within minutes to hours (e.g., Nissen & Bullemer, 1987; Robertson, 2007; Hardwick et al., 2019; Frölich et al., 2023). Previous work shows that such repetition under time pressure yields behavior that is fast, inflexible, and outcome-insensitive, which are, to our knowledge, some defining criteria of habitual control in these frameworks. We have now added a paragraph in the Limitations section explicitly acknowledging that the present task captures one perspective on the mechanistic essence but not the temporal scope or complexity of real-world habits.

On p16 we write:

“It is important to note that the habitual sequence studied here develops over minutes to hours rather than weeks or months. While real-world habits such as making coffee or smoking arise from hundreds of repetitions across diverse contexts, the present paradigm aims to capture a core mechanistic component of habit formation, namely repetition-driven automatization, on an experimentally tractable time scale. Similar short-term sequence-learning paradigms have long been used to study procedural habits (Nissen & Bullemer, 1987; Robertson 2007, Hardwick et al., 2019). Thus, our findings should not be interpreted as modeling long-term behavioral routines but rather as revealing how rapidly acquired habit-like action sequences can be adaptively modulated by context under time pressure. We acknowledge that uncertainty remains regarding the extent to which this paradigm generalizes to real-world habits and that studies with longer time scales (e.g. with gamified smartphone task) should be conducted.”

Reviewer #3:

Third, related to the previous point, the authors report findings that align with the hypothesis of distinct neural systems (dorsal vs. ventral striatum) for habits and goal-directed actions. However, what is the actual evidence for this dissociation in humans? I could not find convincing, recent evidence supporting this claim among the citations provided. I do not believe that these neural hypotheses can be considered well-supported or consensual. The authors should clarify the empirical basis for this assumption.

Response:

We thank the reviewer for this comment and agree that a strict ventral to dorsal striatal dissociation between habitual and goal-directed control is no longer considered state of the art. Current thinking seems to converge on a functional gradient within the striatum (Kelly, 2004; Gremel et al., 2013; Corbit, 2018; Kato et al., 2025), such that reward- and goal-directed processes are more strongly associated with orbitofrontal and ventral striatal regions, whereas action selection and therefore repetition-related processes are more closely linked to medial and dorsal striatal activity. Importantly, these functional specializations are embedded within complex, spiraling, and interacting corticostriatal circuits (Haber et al., 2001; Haber 2016). Consistent with this view, recent work suggests that habitual and goal-directed processing are neurally intertwined rather than cleanly separable (Kono et al., 2023), supporting models that emphasize overlapping and interacting corticostriatal circuits (Haber, 2016), as well as arbitration and control mechanisms (Bender et al., 2024; Kono et al., 2023; Cataldi et al., 2024), rather than a clear separation. We believe that our findings are still compatible with such an overlapping and interacting mechanism.

We now acknowledge this in the introduction and the discussion accordingly:

p2 in the introduction:

“However, recent work suggests that habitual and goal-directed processing are neurally intertwined rather than cleanly separable (e.g. Kono et al., 2023), supporting models that emphasize overlapping and interacting corticostriatal circuits (Haber, 2016) as well as arbitration and control mechanisms (Bender et al., 2023; Kono et al., 2023; Cataldi et al., 2024), rather than clear separation.”

and p15 in the discussion:

“

Although we did not collect neural data in the present study, our findings are in principle compatible with evidence for overlapping and interacting corticostriatal circuits (Haber, 2016) as well as arbitration and control mechanisms (Bender et al., 2023; Kono et al., 2023; Cataldi et al., 2024) that flexibly adjust the influence of a more deliberate goal-directed or simpler stimulus-response control over behavioral output.

“

Reviewer #3:

Fourth, the authors propose that arbitration via context detection is computationally less costly than the “traditional” method. However, this claim depends on a clear estimate of the cost of context detection itself. In the current task, context detection is trivial and effectively built into the model as an observable variable. But what about real-life situations? Is context detection always easier? This remains unclear to me. I would like to see the authors provide more justification and discussion on this point.

Response: We thank the reviewer for bringing that up and feel that this is an important point that warrants discussion.

We added a paragraph on p15 in the Discussion:

“As outlined in the introduction, we argued that contextual inference can be computationally appealing because it enables fast and flexible adaptation. This perspective builds on work in motor control (Heald et al., 2021; Cuevas Rivera & Kiebel, 2023), learning and memory retrieval (Heald et al., 2023), and decision-making under uncertainty (Schwöbel et al., 2021, Schwöbel et al., 2024), and fits well with our paradigm, in which context detection is presumably comparatively cheap. We emphasize that this claim is task-dependent and does not necessarily generalize to all real-world settings. When contexts are few, stable, and/or signaled by reliable cues, inferring contexts can reduce policy complexity and expected computational costs by allowing the reuse of context-specific action mappings. In contrast, when contexts are numerous, rapidly changing, or weakly signaled, context inference itself may become costly and outweigh these benefits. From a resource-rational perspective, the computational efficiency of contextual inference therefore depends on environmental structure, learning history and expected outcomes.”

Reviewer #3

Fifth and finally, I believe the authors are overlooking an important body of literature on dual-controller systems, particularly regarding the effects of time pressure on performance. In this respect, they should consider the work of Wim De Neys and his reviews. In my view, their task is strongly connected to the system 1/system 2 framework, which seems more plausible than the concept of habit formation given the short time frame of their study.

Response: We thank the reviewer for pointing out the close connection between our findings and the dual-process and conflict-monitoring literature. We agree that our task strongly engages rapid conflict detection under time pressure, and we now explicitly discuss this relation in a new paragraph in the revised Discussion on p15:

“A related perspective comes from the dual-process literature on conflict detection under time pressure, particularly the work by De Neys and colleagues. Here the idea is that when responses are generated rapidly and intuitively, conflicting response tendencies are often detected automatically, leading to measurable slowing and performance costs, without invoking slow, reflective deliberation (De Neys et al., 2008; De Neys & Wim, 2012; Evans & Stanovich, 2013; De Neys & Wim 2018). Crucially, such conflict detection has been shown to operate under severe time constraints comparable to those in our task. From this perspective, the congruency effects observed does not necessarily imply deliberative “System 2” intervention. Instead, they are compatible with a fast, automatic conflict-monitoring mechanism that detects misalignment between a prepotent response tendency (here, the habitual action sequence) and current task goals (reward maximization). In our computational framework, this conflict detection is expressed as a context-dependent modulation of the influence of repetition-based habit values on choice, rather than as a separate deliberative controller. Thus, from one perspective, the habit–goal interaction studied here can be understood as a computationally explicit instantiation of the fast conflict-detection processes emphasized in contemporary dual-process accounts. On the other hand, the work of De Neys and colleagues has primarily focused on belief- and judgment-based decisions e.g., \cite{de2008conflict,de2023advancing} rather than habit-like motor action sequences or reward maximization. Accordingly, further research is warranted to clarify the overlap and/or domain-specific differences between both lines of work.”

Response to the Editor and Reviewers for Wagner & Frölich et al., “Context-Dependent Interaction Between Goal-Directed and Habitual Control Under Time Pressure” in Communications Psychology

Manuscript ID: COMMSPSYCHOL-25-0008-A

General remarks:

We thank the Editor and the Reviewers for their positive evaluation of our manuscript and their helpful and constructive feedback. We have revised the manuscript one last time to address the remain point regarding the exclusion of 5 participants. Furthermore, while implementing the Editor's request to include confidence intervals for all behavioral analyses, we discovered a discrepancy in the participant samples used across different sections of the manuscript (behavioral analyses, RL models, DDM). Originally, to achieve a balanced design, 60 participants were pseudo-randomly selected from our full eligible sample of 65. We realized that, due to separate data-processing scripts, the subset of 60 participants used for the behavioral analyses differed slightly from the subset used for the RL and DDM modeling. To ensure complete consistency, we have synchronized the datasets and re-run all analyses. Consequently, the exact t -values, correlation coefficients, and effect sizes in the revised manuscript have shifted slightly. All RL models are now fitted using MCMC in JAGS (Gibbs-Sampling) rather than Variational Inference therefore providing a direct estimation of the full posterior distributions. Furthermore, this allowed us to compute our individual-level model comparisons using the Expected Log Predictive Density (Δ ELPD) derived from the WAIC, removing reliance on the ELBO to approximate Bayes factors. Given these changes now only 30 -instead of 37- participants are best fitted (BFs > 3) by model 3 compared to model 1. Other than that, none of the results changed in a qualitative way and all original conclusions remain supported. We now report BF's for model comparison and WAIC scores for the group level and for each subject in the Supplementary Information. Finally, for transparency, we now also report behavioral and RL analyses (group level WAIC) and hyperparameter distributions for the complete sample of 65 participants in the Supplementary Information. We also provide both datasets with 60 and 65 participants in our data repository.

Reviewer #1 (Remarks to the Author):

The authors have addressed most of my concerns but one remains as the authors failed to address this in the revision: The authors state that 60 of the eligible 65 participants were analyzed in the present study. How were the 5 participants chosen for exclusion? This seems arbitrary. How exclusions were made should be fully explained in the text.

Response: We thank the reviewer for their positive evaluation of our manuscript and for their helpful feedback. To address this remaining concern, we have further clarified in the Methods section exactly why we planned a balanced sample of participants, and how the 5 participants were chosen for exclusion (i.e., through random selection to match the $n=15$ size of the smallest group (task version)). Furthermore, for transparency, we now report all main analyses (linear inference statistics and RL modeling) for the full, unselected sample of 65 participants in the Supplementary Information. Across these supplementary analyses, all results remain qualitatively unchanged.

Reviewer #2 (Remarks to the Author):

Reviewer #2 (Remarks to the Author):

The authors have done an excellent job revising the manuscript. In particular, the addition of model validation, parameter recovery, and control analyses addressing potential confounding effects has substantially strengthened the work. These revisions have improved the data analyses, enhanced the clarity of the figures, and overall increased the quality of the manuscript. While I am generally convinced that the manuscript presents well-supported findings, it remains unclear why reported results are based on the random exclusion of five participants rather than presenting the analyses for the complete sample. In their responses to Reviewers 1 and 2, the authors state that “the qualitative results, including the model ranking, remain unchanged.” Given this clarification, I recommend that the full-sample analyses be included in the Supplementary Materials to ensure transparency and to allow readers to evaluate the results based on the complete dataset.

Response: We very much thank the reviewer for their positive evaluation of our manuscript and for their continued helpful suggestions. Following your recommendation, we now report the full-sample analyses in the Supplementary Information, including the hyperparameter distributions, model comparisons, and behavioral results.

Reviewer #3 (Remarks to the Author):

The authors did a satisfactory work. There are some typos in the citations, I believe (Neys & Wim ?)

Response: We very much thank the reviewer for their helpful suggestions and support during the revision.